# VideoRoPE: What Makes for Good Video Rotary Position Embedding?

Xilin Wei [* 1 2]  Xiaoran liu [* 1 2 3]  Yuhang Zang [2]  Xiaoyi Dong [2 4]  Pan Zhang [2]  Yuhang Cao [2]  Jian Tong [2]
Haodong Duan [2]  Qipeng Guo [2]  Jiaqi Wang [2]  Xipeng Qiu [1 2 3]  Dahua Lin [2 4 5]

## Abstract

While Rotary Position Embedding (RoPE) and its variants are widely adopted for their long-context capabilities, the extension of the 1D RoPE to video, with its complex spatio-temporal structure, remains an open challenge. This work first introduces a comprehensive analysis that identifies four key characteristics essential for the effective adaptation of RoPE to video, which have not been fully considered in prior work. As part of our analysis, we introduce a challenging V-NIAH-D (Visual Needle-In-A-Haystack with Distractors) task, which adds periodic distractors into V-NIAH. The V-NIAH-D task demonstrates that previous RoPE variants, lacking appropriate temporal dimension allocation, are easily misled by distractors. Based on our analysis, we introduce **VideoRoPE**, with a *3D structure* designed to preserve spatio-temporal relationships. VideoRoPE features *low-frequency temporal allocation* to mitigate periodic oscillations, a *diagonal layout* to maintain spatial symmetry, and *adjustable temporal spacing* to decouple temporal and spatial indexing. VideoRoPE consistently surpasses previous RoPE variants, across diverse downstream tasks such as long video retrieval, video understanding, and video hallucination. Our code is available at https://github.com/Wiselnn570/VideoRoPE.

## 1. Introduction

Rotary Position Embedding (RoPE) (Su et al., 2024) helps Transformer models understand word order by assigning each token a unique positional 'marker' calculated using a mathematical rotation matrix. RoPE has advantages in long-

Table 1. Comparison between different RoPE variants for Video Large Language Models (Video LLMs).

| | 2D/3D Structure | Frequency Allocation | Spatial Symmetry | Temporal Index Scaling |
|---|---|---|---|---|
| Vanilla RoPE (Su et al., 2024) | ✗ | ✗ | ✗ | ✗ |
| TAD-RoPE (Gao et al., 2024) | ✗ | ✗ | ✗ | ✓ |
| RoPE-Tie (Su, 2024a) | ✓ | ✗ | ✓ | ✗ |
| M-RoPE (Wang et al., 2024a) | ✓ | ✗ | ✗ | ✗ |
| VideoRoPE (Ours) | ✓ | ✓ | ✓ | ✓ |

Figure 1. VideoRoPE outperforms RoPE variants on benchmarks.

context understanding (Ding et al., 2024b), and continues to be a default choice in leading Large Language Models (LLMs) like the LLaMA (Touvron et al., 2023a;b; Dubey et al., 2024) and QWen (Yang et al., 2024a;b) series.

The original RoPE implementation (Vanilla RoPE) (Su et al., 2024) is designed for sequential 1D data like text. However, recent Video Large Language Models (Video LLMs) (Li et al., 2023; Lin et al., 2023a; Chen et al., 2024a; Maaz et al., 2024b; Zhang et al., 2024d; Wang et al., 2024c; Chen et al., 2024b; Zhang et al., 2024b) process video, which has a more complex spatio and temporal structure. As shown in Tab. 1, although several RoPE-based approaches (Gao et al., 2024; Wang et al., 2024a) have been proposed to support video inputs, these variants exhibit limitations and do not fully satisfy the following key characteristics:

**(1) 2D/3D Structure.** Some existing Video LLMs direct flatten the video frame into 1D embeddings and apply the 1D structure RoPE (Su et al., 2024; Gao et al., 2024). These solutions fail to capture video data's inherent 2D or 3D (temporal ($t$), horizontal ($x$), and vertical ($y$)) structure, thus hindering explicit spatial and temporal representation.

---
[*]Equal contribution [1]Fudan University, Shanghai, China [2]Shanghai AI Laboratory, Shanghai, China [3]Shanghai Innovation Institute, Shanghai, China [4]The Chinese University of Hong Kong [5]CPII under InnoHK. Correspondence to: Yuhang Zang <zangyuhang@pjlab.org.cn>, Qipeng Guo <guoqipeng@pjlab.org.cn>, Jiaqi Wang <wangjiaqi@pjlab.org.cn>.

*Proceedings of the 42$^{nd}$ International Conference on Machine Learning*, Vancouver, Canada. PMLR 267, 2025. Copyright 2025 by the author(s).

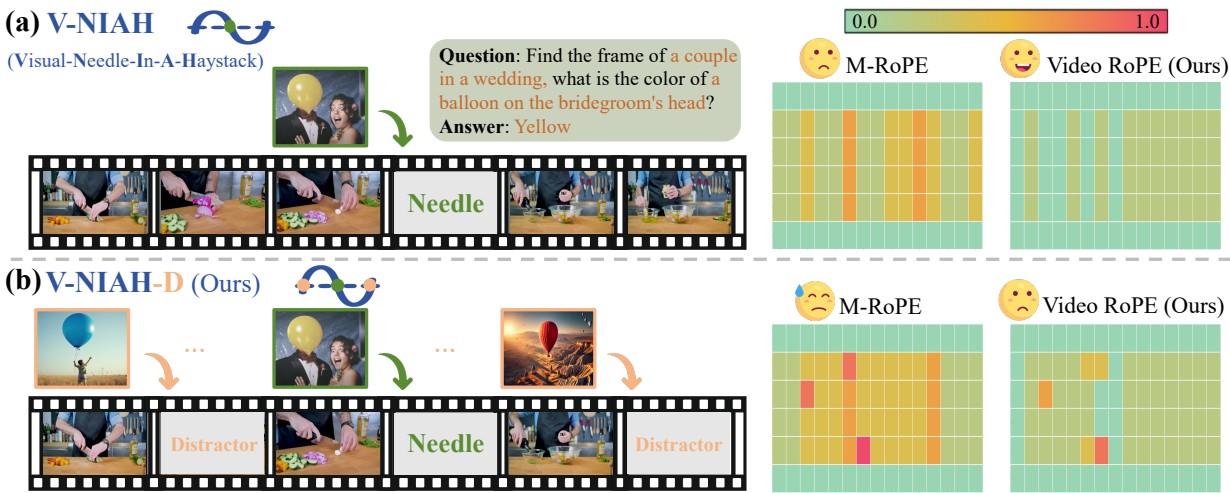

*Figure 2.* **Left:** To demonstrate the importance of frequential allocation, based on VIAH (**a**) we present a more challenging V-NIAH-D task (**b**) that similar images are inserted as distractors. **Right:** Compared to M-RoPE, our VideoRoPE is more robust in retrieval and is less affected by distractors. See Fig. 7 in the Experiments section for details on the horizontal and vertical axes.

**(2) Frequency Allocation.** Previous approaches such as M-RoPE used in QWen2-VL (Wang et al., 2024a) employ 3D structure, dividing feature dimensions into distinct subsets for $(t, x, y)$ encoding, respectively. How to determine the optimal allocation of these dimension subsets, and their associated frequencies [1] are not well studied. Some previous work allocates the lower dimensions corresponding to the high frequency to represent the $t$. However, the temporal dimension $t$ is significantly tortured by periodic oscillation, and distant positions may have the same embeddings.

We present a simple setting to verify this point. Based on the previous long-video retrieval task V-NIAH (Visual Needle-In-A-Haystack) (Zhang et al., 2024d), we insert several similar images that do not affect the question's answer before and after the needle image as distractor (Hsieh et al., 2024; Yuan et al., 2024), forming a new task, V-NIAH-D (Visual Needle-In-A-Haystack with Distractors). As shown in Fig. 2, we find that previous M-RoPE is misled by distractors, showing a significant performance decline from V-NIAH to V-NIAH-D. Our observation demonstrates that the periodic oscillation reduces Video LLMs' robustness.

**(3) Spatial Symmetry.** The distance between the end of the precedent textual input and the start of visual input equals the distance between the end of visual input and the start of subsequent textual input (Su, 2024b). Such a symmetry ensures that the visual input receives equal contextual influence from both the preceding and subsequent textual information.

**(4) Temporal Index Scaling.** Spatial and temporal dimensions often exhibit different granularities (e.g., a unit change in $x/y$ differs from a unit change in $t$) (Gao et al., 2024). Employing varying index intervals in positional encoding allows for dimension-specific encoding, capturing diverse scales and enhancing efficiency.

Driven by our analysis, we present a new video position embedding strategy, **VideoRoPE**, which can simultaneously satisfy the four properties in Tab. 1. Specifically, we use a 3D structure to model spatiotemporal information, allocating higher dimensions (lower frequencies), to the temporal axis (**L**ow-frequency **T**emporal **A**llocation, **LTA**) to prioritize temporal modeling. The right panel of Fig. 2 demonstrates that our LTA allocation mitigates oscillations and exhibits robustness to distractors in the V-NIAH-D task. We further employ a **D**iagonal **L**ayout (**DL**) design to ensure spatial symmetry and preserve the relative positioning between visual and text tokens. Regarding temporal index scaling, we propose **A**djustable **T**emporal **S**pacing (**ATS**), where a hyper-parameter controls the relative temporal spacing of adjacent visual tokens. In summary, our proposed position encoding scheme demonstrates favorable characteristics for modeling video data, yielding a robust and effective representation of positional information.

Overall, the contributions of this work are summarized as:

**(1)** We present an analysis of four key properties essential for RoPE when applied to video. Motivated by this analysis, we propose VideoRoPE including Low-frequency Temporal Allocation (LTA), Diagonal Layout (DL), and Adjustable Temporal Spacing (ATS) to satisfy all four properties.

**(2)** We introduce the challenging V-NIAH-D task to expose the drawbacks of current position embedding designs regarding frequency allocation. We reveal that existing Video

---

[1] In RoPE, frequencies are determined by $\beta^{-2n/d}$, where $\beta$ is a constant, $n$ is the dimension index, $d$ is the total number of dimensions. Thus, choosing which dimensions represent $t$, $x$, and $y$ directly determines the frequencies used for each.

LLMs are easily misled to frequency-based distractors.

**(3)** Extensive experiments demonstrate that VideoRoPE consistently achieves superior performance compared to other RoPE variants. For example, VideoRoPE outperforms previous M-RoPE on long video retrieval (**+12.4** on V-NIAH, **+12.4** on V-NIAH-D), video understanding (**+2.9** on LongVideoBench, **+4.5** on MLVU, **+1.7** on Video-MME) and hallucination (**+11.9** on VideoHallucer) benchmarks.

## 2. Related Work

**RoPE (Rotary Position Embedding).** RoPE (Su et al., 2024) is a pivotal mechanism for encoding positional information in LLM long-context modeling. Using a rotation matrix, RoPE unifies the advantages of both absolute and relative positional embedding schemes. In RoPE design, different feature dimensions are embedded with position information based on Trigonometric functions $\sin$ and $\cos$ with different frequencies (Peng et al., 2023; Liu et al., 2023b). Lower dimensions correspond to higher frequency given larger values of base frequency. The simplicity and effectiveness of RoPE have led to its widespread adoption in leading LLMs (Touvron et al., 2023a; Yang et al., 2024a; Team et al., 2024; Cai et al., 2024; Sun et al., 2024).

**Extending RoPE to Multi-Modal Data.** Extending RoPE to multi-modal or Video LLMs typically follows two approaches. One approach directly applies standard RoPE, flattening visual tokens and treating text and visual tokens as a single 1D sequence. Although variants (e.g., TAD-RoPE (Gao et al., 2024)) introduce enhancements in indexing and attention mechanisms, these 1D RoPE variants overlook the spatiotemporal structure of video and inherent inter-modal differences (Su, 2024a;b; Wang et al., 2024a). In contrast, several studies have explored incorporating structural information to formulate the 2D/3D RoPE. For example, some previous works (Agrawal et al., 2024; Wang et al., 2024a) integrate RoPE-2D into visual encoders to improve spatial representation, particularly for resolution scaling. Based on the RoPE-Tie (Su, 2024a), M-RoPE (Wang et al., 2024a) used in QWen2-VL further generalizes RoPE to three dimensions to model both temporal and spatial dynamics. While effective, M-RoPE exhibits limitations, such as struggles with distractors in our V-NIAH-D task. This work presents a comprehensive analysis of the important characteristics essential for extending RoPE to video and proposes VideoRoPE according to our analysis.

## 3. Analysis

**3D Structure.** The vanilla RoPE defines a matrix $\boldsymbol{A}_{t_1,t_2}$ that represents the relative positional encoding between two positions $t_1$ and $t_2$ in a 1D sequence:

$$\boldsymbol{A}_{t_1,t_2} = \left(\boldsymbol{q}_{t_1}\boldsymbol{R}_{t_1}\right)\left(\boldsymbol{k}_{t_2}\boldsymbol{R}_{t_2}\right)^\top = \boldsymbol{q}_{t_1}\boldsymbol{R}_{\Delta t}\boldsymbol{k}_{t_2}^\top, \quad (1)$$

where $\Delta t = t_1 - t_2$, the symbols $\boldsymbol{q}_{t_1}$ and $\boldsymbol{k}_{t_2}$ are the query and key vectors at positions $t_1$ and $t_2$. The *relative rotation matrix* $\boldsymbol{R}_{\Delta t}$ is defined as $\boldsymbol{R}_{\Delta t} = \exp(\Delta t i\theta_n)$, while $i$ is the imaginary unit, $\theta_n = \beta^{-2n/d}$ is the frequency of rotation applied to a specific $n$-th pair of $d$ dimensions ($n = 0, \ldots, d/2 - 1$), and $\beta$ is the frequency base parameter. The vanilla RoPE uses $d = 128$, thus $n = 0, \ldots, 63$. Consequently, the $\boldsymbol{A}_{t_1,t_2}$ in Eq. (1) can be extended as:

$$\begin{pmatrix} q^{(0)} \\ q^{(1)} \\ \vdots \\ q^{(126)} \\ q^{(127)} \end{pmatrix}^\top \begin{pmatrix} \cos\theta_0\Delta t & -\sin\theta_0\Delta t & \cdots & 0 & 0 \\ \sin\theta_0\Delta t & \cos\theta_0\Delta t & \cdots & 0 & 0 \\ \vdots & \vdots & \ddots & \vdots & \vdots \\ 0 & 0 & \cdots & \cos\theta_{63}\Delta t & \sin\theta_{63}\Delta t \\ 0 & 0 & \cdots & \sin\theta_{63}\Delta t & \cos\theta_{63}\Delta t \end{pmatrix} \begin{pmatrix} k^{(0)} \\ k^{(1)} \\ \vdots \\ k^{(126)} \\ k^{(127)} \end{pmatrix} \quad (2)$$

While the vanilla RoPE operates on 1D sequences, it can also be applied to higher-dimensional input by flattening the input into a 1-D sequence. However, the flattening process discards crucial neighborhood information, increases the sequence length, and hinders the capture of long-range dependencies. Therefore, preserving the inherent 3D structure is essential when adapting RoPE for video data. Some recent RoPE-variants (e.g., M-RoPE in Qwen2-VL (Wang et al., 2024a)) incorporate the 3D structure. The corresponding relative matrix $\boldsymbol{A}_{(t_1,x_1,y_1)}$ is computed as:

$$\boldsymbol{A}_{(t_1,x_1,y_1),(t_2,x_2,y_2)} = \boldsymbol{q}_{(t_1,x_1,y_1)}\boldsymbol{R}_{\Delta t,\Delta x,\Delta y}\boldsymbol{k}_{(t_2,x_2,y_2)}^\top, \quad (3)$$

where $\Delta t = t_1 - t_2$, $\Delta x = x_1 - x_2$, and $\Delta y = y_1 - y_2$. M-RoPE divides the $d = 128$ feature dimensions into 3 groups: the first 32 for temporal positions ($t$), the middle 48 for horizontal positions ($x$), and the last 48 for vertical positions ($y$). As shown in Eq (4), $\boldsymbol{A}_{(t_1,x_1,y_1),(t_2,x_2,y_2)}$ in M-RoPE is extended as:

$$\begin{pmatrix} q^{(0)} \\ q^{(1)} \\ q^{(2)} \\ q^{(3)} \\ \vdots \\ q^{(30)} \\ q^{(31)} \end{pmatrix}^\top \begin{pmatrix} \cos\theta_0\Delta t & -\sin\theta_0\Delta t & 0 & 0 & \cdots & 0 & 0 \\ \sin\theta_0\Delta t & \cos\theta_0\Delta t & 0 & 0 & \cdots & 0 & 0 \\ 0 & 0 & \cos\theta_1\Delta t & -\sin\theta_1\Delta t & \cdots & 0 & 0 \\ 0 & 0 & \sin\theta_1\Delta t & \cos\theta_1\Delta t & \cdots & 0 & 0 \\ \vdots & \vdots & \vdots & \vdots & \ddots & \vdots & \vdots \\ 0 & 0 & 0 & 0 & \cdots & \cos\theta_{15}\Delta t & -\sin\theta_{15}\Delta t \\ 0 & 0 & 0 & 0 & \cdots & \sin\theta_{15}\Delta t & \cos\theta_{15}\Delta t \end{pmatrix} \begin{pmatrix} k^{(0)} \\ k^{(1)} \\ k^{(2)} \\ k^{(3)} \\ \vdots \\ k^{(30)} \\ k^{(31)} \end{pmatrix}$$

$$\underbrace{\phantom{xxxxxxxxxxxxxxxxxxxxxxxxxxxxx}}_{\text{modeling temporal dependency with higher frequency}}$$

$$+\begin{pmatrix} q^{(32)} \\ q^{(33)} \\ q^{(34)} \\ q^{(35)} \\ \vdots \\ q^{(78)} \\ q^{(79)} \end{pmatrix}^\top \begin{pmatrix} \cos\theta_{16}\Delta x & -\sin\theta_{16}\Delta x & 0 & 0 & \cdots & 0 & 0 \\ \sin\theta_{16}\Delta x & \cos\theta_{16}\Delta x & 0 & 0 & \cdots & 0 & 0 \\ 0 & 0 & \cos\theta_{17}\Delta x & -\sin\theta_{17}\Delta x & \cdots & 0 & 0 \\ 0 & 0 & \sin\theta_{17}\Delta x & \cos\theta_{17}\Delta x & \cdots & 0 & 0 \\ \vdots & \vdots & \vdots & \vdots & \ddots & \vdots & \vdots \\ 0 & 0 & 0 & 0 & \cdots & \cos\theta_{39}\Delta x & -\sin\theta_{39}\Delta x \\ 0 & 0 & 0 & 0 & \cdots & \sin\theta_{39}\Delta x & \cos\theta_{39}\Delta x \end{pmatrix} \begin{pmatrix} k^{(32)} \\ k^{(33)} \\ k^{(34)} \\ k^{(35)} \\ \vdots \\ k^{(78)} \\ k^{(79)} \end{pmatrix}$$

$$\underbrace{\phantom{xxxxxxxxxxxxxxxxxxxxxxxxxxxxx}}_{\text{modeling horizontal dependency with intermediate frequency}}$$

$$+\begin{pmatrix} q^{(80)} \\ q^{(81)} \\ q^{(82)} \\ q^{(83)} \\ \vdots \\ q^{(126)} \\ q^{(127)} \end{pmatrix}^\top \begin{pmatrix} \cos\theta_{40}\Delta y & -\sin\theta_{40}\Delta y & 0 & 0 & \cdots & 0 & 0 \\ \sin\theta_{40}\Delta y & \cos\theta_{40}\Delta y & 0 & 0 & \cdots & 0 & 0 \\ 0 & 0 & \cos\theta_{41}\Delta y & -\sin\theta_{41}\Delta y & \cdots & 0 & 0 \\ 0 & 0 & \sin\theta_{41}\Delta y & \cos\theta_{41}\Delta y & \cdots & 0 & 0 \\ \vdots & \vdots & \vdots & \vdots & \ddots & \vdots & \vdots \\ 0 & 0 & 0 & 0 & \cdots & \cos\theta_{63}\Delta y & -\sin\theta_{63}\Delta y \\ 0 & 0 & 0 & 0 & \cdots & \sin\theta_{63}\Delta y & \cos\theta_{63}\Delta y \end{pmatrix} \begin{pmatrix} k^{(80)} \\ k^{(81)} \\ k^{(82)} \\ k^{(83)} \\ \vdots \\ k^{(126)} \\ k^{(127)} \end{pmatrix}$$

$$\underbrace{\phantom{xxxxxxxxxxxxxxxxxxxxxxxxxxxxx}}_{\text{modeling vertical dependency with lower frequency}}$$

$$(4)$$

**Frequency Allocation.** Incorporating 3D structure raises the question of how to allocate the temporal ($t$), horizontal ($x$), and vertical ($y$) components within the $d$ dimensions. Note that different allocation strategies are not equivalent

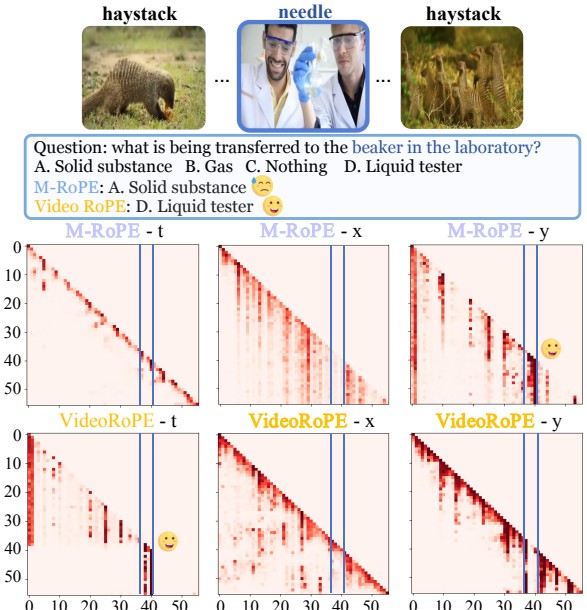

*Figure 3.* Attention-based frequential allocation analysis. **Middle**: M-RoPE's temporal dimension ($t$) is limited to local information, resulting in a diagonal layout. **Bottom**: VideoRoPE effectively retrieves the needle using the temporal dimension. The x and y coordinates represent the video frame number, e.g., 50 for 50 frames. For more details see Appendix E.

in the rotation frequency $\theta_n = \beta^{-2n/d}$. As shown in Eq. (4), M-RoPE assigns higher frequencies (corresponding to lower dimensions) to the temporal dimension ($t$).

To highlight the importance of frequency allocation, we introduce a challenging retrieval task **V**isual **N**eedle-**I**n-**A**-**H**astack-**D**istractor (**V-NIAH-D**). V-NIAH-D builds upon V-NIAH (Zhang et al., 2024d), a benchmark designed to evaluate visual long-context understanding. However, the straightforward retrieval-based task has been shown to provide only a superficial form of long-context understanding (Hsieh et al., 2024; Yuan et al., 2024). Therefore, We enhance V-NIAH by incorporating semantically similar distractors, obtained using Google Image Search (Google, 2025) or Flux (Labs, 2023), to mitigate the possibility of correct answers through random chance. These distractors are designed to be unambiguous to the question in Fig. 2.

As shown in Fig. 2, M-RoPE exhibits a clear performance drop from V-NIAH to V-NIAH-D. To investigate this decline, we follow previous works (Xiao et al., 2023; Liu et al., 2023b; Barbero et al., 2024) to visualize the attention scores in Fig. 3. We decompose the attention scores into their corresponding temporal ($t$), horizontal ($x$), and vertical ($y$) components for visualization.

Fig. 3 reveals unusual M-RoPE's attention patterns, despite locating the needle image, it fails to answer the multi-choice question. According to M-RoPE's attention, the needle is

located primarily through vertical positional information, rather than temporal features. Thus, the temporal dimension fails to capture long-range semantic dependencies, focusing on local relationships. Conversely, the spatial dimensions capture long-range rather than local semantic information. Lastly, the horizontal and vertical dimensions display distinct characteristics, with the vertical dimension exhibiting phenomena reminiscent of attention sinks (Xiao et al., 2023). These suggest the performance decline primarily results from sub-optimal frequency allocation designs of M-RoPE.

**Spatial Symmetry.** Given the text tokens $T$ and the visual tokens $T_v$, spatial symmetry (Su, 2024b) claims that the distance between the end of the preceding textual input ($T_{\text{pre}}$) and the beginning of the visual input ($T_v^{\text{start}}$) is equal to the distance between the end of the visual input ($T_v^{\text{end}}$) and the beginning of the subsequent textual input ($T_{\text{sub}}$):

$$T_v^{\text{start}} - T_{\text{pre}} = T_{\text{sub}} - T_v^{\text{end}}. \tag{5}$$

The spatial symmetrical structure can potentially simplify the learning process and reduce bias toward input order. However, existing 3D RoPE variants such as M-RoPE do not meet the spatial symmetry, we will elaborate related discussion in Fig. 6.

**Temporal Index Scaling.** The frame index in video and the token index in text are inherently different (Su, 2024b; Li et al., 2024a). Recognizing this difference, methods like TAD-RoPE, a 1D RoPE adaptation for Video LLMs, introduce distinct step offsets for image and text token indices: $\gamma$ for image tokens and $\gamma + 1$ for text tokens. Consequently, an ideal RoPE design for video data should permit scaling of the temporal index to meet the inherent difference between the frame index and the text index.

## 4. VideoRoPE

Based on some previous research and the above analysis, we claim that a good RoPE design for Video LLMs, especially for long videos, should satisfy four requirements. The first requirement has been solved by RoPE-Tie (Su, 2024a) and the subsequent M-RoPE (Wang et al., 2024a). To solve the last three requirements and mitigate the performance decline observed in V-NIAH-D, we propose our VideoRoPE, comprising the following three key components.

**Low-frequency Temporal Allocation (LTA).** As shown in Eq. (2), the vanilla RoPE (Su et al., 2024) uses all dimensions to model the 1D position information. And as indicated in Eq. (4), M-RoPE (Wang et al., 2024a) uses different dimensions to model temporal, horizontal, and vertical dimensions sequentially. However, previous frequency allocation strategies are suboptimal because different RoPE dimensions capture dependencies at varying ranges. As shown in Fig. 3, an interesting observation is that the local

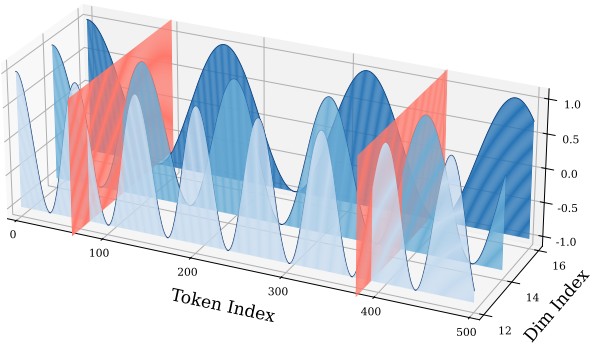

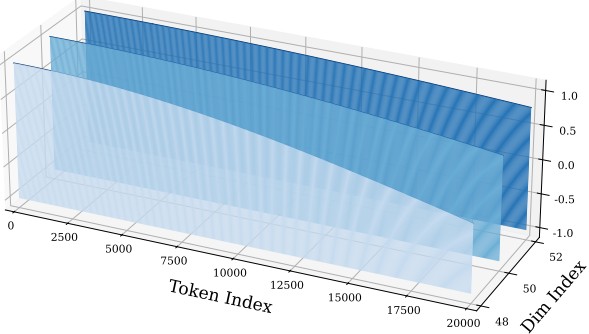

(a) Temporal Frequency Allocation in M-RoPE

(b) Temporal Frequency Allocation in VideoRoPE (ours)

*Figure 4.* **(a)** M-RoPE (Wang et al., 2024a) models temporal dependencies using the *first* 16 rotary angles, which exhibit higher frequencies and more pronounced oscillations. **(b)** In contrast, VideoRoPE models temporal dependencies using the *last* 16 rotary angles, characterized by significantly wider, monotonic intervals. Our frequency allocation effectively mitigates the misleading influence of distractors in V-NIAH-D. For a more detailed analysis, please refer to Appendix F.

attention branch (as reported in (Han et al., 2024)) corresponds to lower dimensions, while the global branch (or attention sink, as in (Xiao et al., 2023)) corresponds to higher dimensions. To sum up, lower dimensions (higher frequency, shorter monotonic intervals, larger $\theta_n$) tend to capture relative distances and local semantics (Men et al., 2024; Barbero et al., 2024), while higher dimensions (lower frequency, wider monotonic intervals, smaller $\theta_n$) capture longer-range dependencies (Barbero et al., 2024).

Based on our analysis, VideoRoPE uses higher dimensions for temporal features in longer contexts and lower dimensions for spatial features, which are limited by resolution and have a fixed range. To avoid the gap between horizontal and vertical positions, we interleave the dimensions responsible for these spatial features. The dimension distribution for VideoRoPE is shown in Eq. (6):

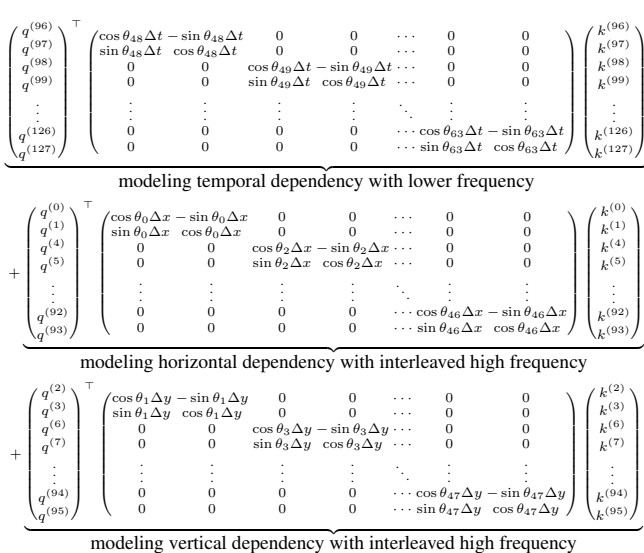

The horizontal position $x$ and vertical position $y$ are inter-

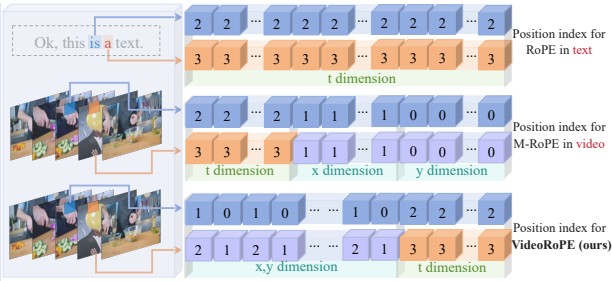

*Figure 5.* The position embeddings of adjacent text tokens for Vanilla RoPE (**top** row), the corresponding visual tokens in adjacent frames for M-RoPE (**middle** row) and our VideoRoPE (**bottom** row) with interleaved spatial and temporal last design.

leaved to occupy the lower dimensions, followed by temporal $t$, which occupies the higher dimensions. We keep the same allocation number for $x$, $y$, and $t$ as M-RoPE for a fair comparison, with values of 48, 48, and 32, respectively. The advantages of this distribution are evident in Fig. 4. For a RoPE-based LLM with a 128-dimensional head (64 rotary angles $\theta_n$), we visualize the function of $\cos\theta_n t$ for 3 dimensions using parallel blue planes.

As shown in Fig. 4 (**a**), M-RoPE's temporal position embeddings are significantly distorted by periodic oscillations (Men et al., 2024), leading to identical embeddings for distant positions. For instance, considering the last three rotary angles, the temporal embeddings are severely affected by these oscillations due to their short monotonic intervals (and even shorter intervals in lower dimensions). This periodicity creates "hash collisions" (red planes), where distant positions share near-identical embeddings, making the model susceptible to distractor influence. Fortunately, our VideoRoPE (Fig. 4 (**b**)) is free from oscillation and Hash collision in temporal modeling. The relationship between periodicity, monotonicity, and temporal modeling is visualized in Fig 4.

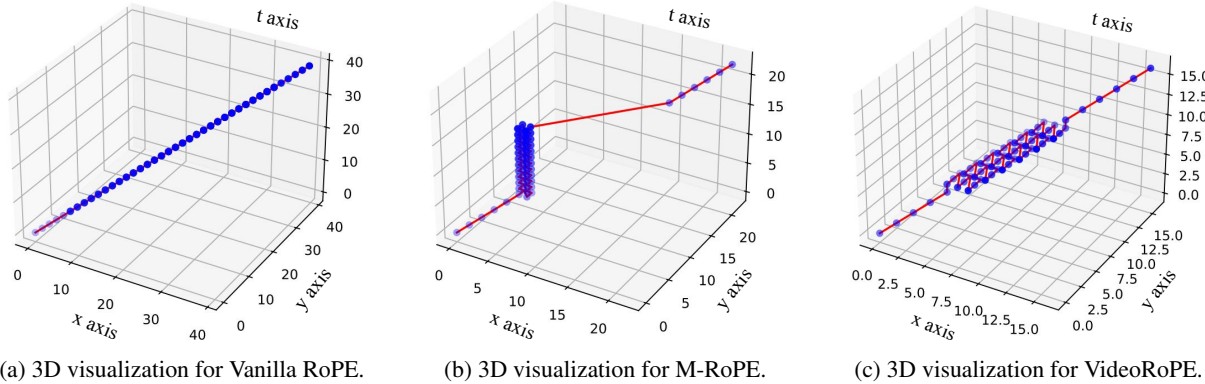

(a) 3D visualization for Vanilla RoPE.  (b) 3D visualization for M-RoPE.  (c) 3D visualization for VideoRoPE.

*Figure 6.* The 3D visualization for different position embedding. **(a)** The vanilla 1D RoPE (Su et al., 2024) does not incorporate spatial modeling. **(b)** M-RoPE (Wang et al., 2024a), while have the 3D structure, introduces a discrepancy in index growth for visual tokens across frames, with some indices remaining constant. **(c)** In contrast, our VideoRoPE achieves the desired balance, maintaining the consistent index growth pattern of vanilla RoPE while simultaneously incorporating spatial modeling.

**Diagonal Layout.** Fig. 6 provides a visual comparison of spatial symmetry in positional encodings. For vanilla RoPE (Fig. 6a), no spatial relation is considered and the index for every dimension increases directly. While M-RoPE (Fig. 6b), incorporates spatial information within each frame, it introduces two significant discontinuities between textual and visual tokens. This arises from M-RoPE's placement strategy, if the first visual token is at $(0, 0)$, the last token in each frame will always be placed at $(W - 1, H - 1)$, creating a stack in the bottom-left corner. Furthermore, like vanilla RoPE, M-RoPE's indices increase with input length across all dimensions.

To address these limitations, VideoRoPE arranges the entire input along the diagonal, see Fig. 6c. The central patch's 3D position for each video frame is $(t, t, t)$, with other patches offset in all directions. Our **Diagonal Layout** has two advantages: (1) our design preserves the relative positions of visual tokens and ensures approximate equidistance from the image corners to the center, preventing text tokens from being overly close to any corner. (2) It maintains the indexing pattern of vanilla RoPE (Fig. 5), as the position index increment between corresponding spatial locations in adjacent frames mirrors that of adjacent textual tokens.

**Adjustable Temporal Spacing.** To scale the temporal index, we introduce a scaling factor $\delta$ to better align temporal information between visual and textual tokens.

Suppose the symbol $\tau$ denotes the token index, for the starting text ($0 \leq \tau < T_s$), the temporal, horizontal, and vertical indices are simply set to the raw token index $\tau$. For the video input ($T_s \leq \tau < T_s + T_v$), The difference $\tau - T_s$ represents the index of the current frame relative to the start of the video, which is then scaled by $\delta$ to control the space in the temporal dimension. For the ending text ($T_s + T_v \leq \tau < T_s + T_v + T_e$), the temporal, horizontal, and vertical index are the same, creating a linear progression.

According to our adjustable temporal spacing design, for a multi-modal input that consists of a text with $T_s$ tokens, a following video with $T_v$ frame with $W \times H$ patches in each frame, and an ending text with $T_e$ tokens, the position indices $(t, x, y)$ of VideoRoPE for $\tau$-th textual token or $(\tau, w, h)$-th visual token are defined as Eq. (7):

$$(t, x, y) = \begin{cases} (\tau, \tau, \tau) & \text{if } 0 \leq \tau < T_s \\ \begin{pmatrix} T_s + \delta(\tau - T_s), \\ T_s + \delta(\tau - T_s) + w - \frac{W}{2}, \\ T_s + \delta(\tau - T_s) + h - \frac{H}{2} \end{pmatrix} & \text{if } T_s \leq \tau < T_s + T_v \\ \begin{pmatrix} \tau + (\delta - 1)T_v, \\ \tau + (\delta - 1)T_v, \\ \tau + (\delta - 1)T_v \end{pmatrix} & \text{if } T_s + T_v \leq \tau < T_s + T_v + T_e \end{cases} \tag{7}$$

where $w$ and $h$ represent the horizontal and vertical indices of the visual patch within the frame, respectively.

In summary, the parameter $\delta$ in our adjustable temporal spacing allows for a flexible and consistent way to encode the relative positions of text and video tokens.

## 5. Experiment

### 5.1. Experimental Setup

**Training Data.** We use a subset of LLaVA-Video-178k dataset (Zhang et al., 2024e) to train VideoRoPE. The LLaVA-Video-178k dataset covers 178k videos and around 5 million question-answers (QA) pairs from diverse sources such as HD-VILA (Xue et al., 2022), Kinetics (Kay et al., 2017), and ActivityNet (Fabian Caba Heilbron & Niebles, 2015). To balance training efficiency and long-video comprehension, we randomly select 136k videos with durations under 2 minutes and 18k videos with durations between 2 and 3 minutes. This process yielded our training set of approximately 1.3 million pairs.

**Implementation Details.** Using the aforementioned video

*Table 2.* **Comparison of different RoPE methods on LongVidionBench, MLVU, and Video-MME**. The benchmarks evaluate performance across three context lengths: 8k, 16k, 32k, and 64k, where **8k** represents context within the training range, and others represent context outside the training range. Our VideoRoPE outperforms other RoPE variants across all three benchmarks. The best results are marked in **bold**, and the second-best results are underlined. For more information on the evaluation, see Appendix B.

| Method | LongVideoBench | | | | MLVU | | | | Video-MME | | | |
|---|---|---|---|---|---|---|---|---|---|---|---|---|
| | 8k | 16k | 32k | 64k | 8k | 16k | 32k | 64k | 8k | 16k | 32k | 64k |
| Vanilla RoPE (Su et al., 2024) | **54.97** | 54.87 | 54.56 | 54.04 | 63.31 | 65.79 | 65.93 | 62.02 | 60.67 | 60.00 | 61.33 | 58.33 |
| TAD-RoPE (Gao et al., 2024) | 54.14 | 55.08 | 53.94 | 53.42 | 63.67 | 65.28 | 65.28 | 60.73 | 60.33 | **61.33** | **62.00** | 58.67 |
| M-RoPE (Wang et al., 2024a) | 53.42 | 52.80 | 53.11 | 54.35 | 60.41 | 60.68 | 61.56 | 61.10 | 60.67 | 59.67 | 61.00 | 59.67 |
| VideoRoPE (Ours) | 54.46 | **55.29** | **57.15** | **57.26** | **65.19** | **66.29** | **66.02** | **65.56** | **61.33** | 61.00 | 61.67 | **61.33** |

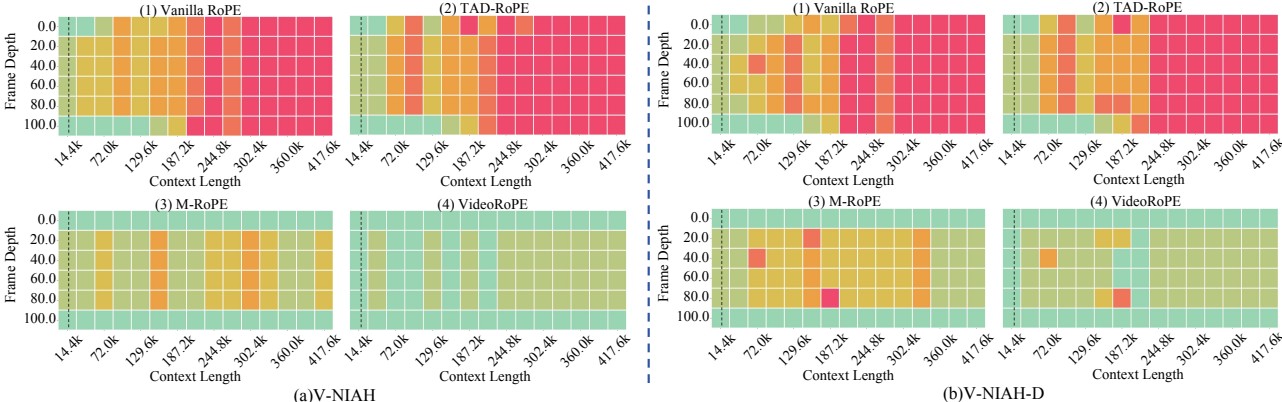

*Figure 7.* Visualization of the retrieval results for V-NIAH and V-NIAH-D. The color gradient from green to red represents the progression of needle retrieval performance, from perfect to zero.

training data, we fine-tune different models that use different positional encoding strategies, such as the Vanilla RoPE (Su et al., 2024), Time-Aware Dual RoPE (TAD-RoPE) (Gao et al., 2024), M-RoPE (Wang et al., 2024a), and our VideoRoPE. All models are initialized with the Vision Transformer from Qwen2-VL-7B and LLM (Vanilla RoPE) from Qwen2-7B (Yang et al., 2024a). Our fine-tuning incorporates our VideoRoPE to process the spatiotemporal nature of the video data effectively. We adopt Qwen2-VL's fine-tuning settings, processing each video at 2 fps with a maximum of 128 frames and dynamically adjusting the image resolution to maintain a consistent token count. However, to prevent memory overflow, we use a context window of 8192 tokens.

Our fine-tuning process employs a batch size of 128, a cosine scheduler with a learning rate of 1e-5, a warm-up ratio of 1e-2, and 704 Nvidia-A100 GPU hours in total. The evaluation involves sampling videos at 2 fps with a minimum of 144 image tokens per frame. We use the vLLM framework (Kwon et al., 2023) to support inference on sequences longer than 32k tokens.

**Evaluation Benchmarks.** We evaluate our approach using six video benchmarks, including tasks related to *long video understanding*, *long video retrieval*, and *video hallucination*. For *long video understanding*, we use **LongVideoBench**

(Wu et al., 2024a) (8 seconds to 1 hour), **MLVU** (Zhou et al., 2024) (3 minutes to 2 hours), and **Video-MME** (Fu et al., 2024) (11 seconds to 60 minutes). For *long video retrieval*, we use **Vision Needle-in-a-Haystack (V-NIAH)** (Zhang et al., 2024d) and our proposed extension, **Vision Needle-in-a-Haystack with Distractors (V-NIAH-D)**, which introduces distractor frames to increase the task difficulty. For *video hallucination*, we use **VideoHallucer** (Wang et al., 2024d), which evaluates the model's ability to correctly answer both basic and hallucinated questions about video content. Details of these benchmarks can be found in Appendix B.

### 5.2. Results on Long Video Understanding

As shown in Tab. 2, we compare our VideoRoPE with existing RoPE variants (vanilla RoPE (Su et al., 2024), TAD-RoPE (Gao et al., 2024), and M-RoPE (Wang et al., 2024a)) across three prominent video understanding benchmarks. Our VideoRoPE consistently outperforms all baseline methods across these benchmarks, demonstrating its robustness and adaptability. Specifically, VideoRoPE achieves improvements of up to 2.91, 4.46, and 1.66 points (64k context length) over the M-RoPE baseline on LongVideoBench, MLVU, and Video-MME, respectively. These results emphasize the superior ability of VideoRoPE to effectively

*Table 3.* **Performance comparison of different RoPEs on V-NIAH and V-NIAH-D**. "Acc." refers to the average accuracy across haystack length and frame depth.

| Method | V-NIAH Acc. | V-NIAH-D Acc. |
|---|---|---|
| Vanilla RoPE (Su et al., 2024) | 31.78 | 30.22 |
| TAD-RoPE (Gao et al., 2024) | 29.33 | 29.56 |
| M-RoPE (Wang et al., 2024a) | 78.67 | 74.67 |
| VideoRoPE | **91.11** | **87.11** |

*Table 4.* **Performance comparison of different RoPEs on VideoHallucer**, evaluated at context lengths of 8k, 16k, 32k, and 64k. The maximum result for each RoPE variant across these context lengths is displayed, with bold for the top result and underlined for the second-highest. 'OR' = Object-Relation, 'T' = Temporal, 'SD' = Semantic Detail, 'F' = Factual, 'NF' = Non-factual.

| Method | OR | T | SD | F | NF | Avg. |
|---|---|---|---|---|---|---|
| Vanilla RoPE (Su et al., 2024) | 51.5 | 30.0 | 48.0 | 8.0 | 43.0 | 36.1 |
| TAD-RoPE (Gao et al., 2024) | 51.0 | 37.0 | 48.0 | 11.5 | 47.5 | 39.0 |
| M-RoPE (Wang et al., 2024a) | 39.0 | 29.0 | 43.5 | 12.5 | 47.5 | 34.3 |
| VideoRoPE | **57.0** | **58.5** | **50.5** | **15.0** | **50.0** | **46.2** |

capture long-range dependencies and maintain performance across various challenging video data tasks.

### 5.3. Results on Long Video Retrieval

Fig. 7 illustrates the performance of V-NIAH and V-NIAH-D with VideoRoPE and other RoPE variants. Specifically, Fig. 7 (a) and (b) demonstrate that the proposed V-NIAH-D is more challenging than V-NIAH. Fig. 7 (1) and (2) show that both Vanilla RoPE and TAD-RoPE exhibit some extrapolation ability beyond the visual training context. However, both methods fail once they exceed a certain extrapolation limit. In contrast, Fig. 7 (3) and (4) highlight the superior performance of VideoRoPE and M-RoPE in extrapolating within the test context range. While both VideoRoPE and M-RoPE successfully handle extrapolation, VideoRoPE consistently outperforms M-RoPE, showcasing the robustness of the task. Tab. 3 provides a quantitative analysis of the retrieval results, demonstrating a 12.44 % performance improvement of our method over M-RoPE on the Video Retrieval task in both settings, confirming the advantages of our proposed method in video retrieval scenarios.

### 5.4. Results on Video Hallucination

As highlighted in Tab. 4, VideoRoPE significantly surpasses current RoPE methods on the VideoHallucer benchmark. In particular, for the Temporal Hallucination task, VideoRoPE demonstrates a substantial performance improvement of 29.5%, indicating its enhanced capability to accurately capture and process temporal dependencies. This improvement suggests that VideoRoPE is better equipped to handle dynamic video sequences, where the understanding of time-based relationships is critical. Similarly, for the Spatial

*Table 5.* **Ablation study about different modules of VideoRoPE.**

| Method | LongVideoBench | | | | MLVU | | | |
|---|---|---|---|---|---|---|---|---|
| | 8k | 16k | 32k | 64k | 8k | 16k | 32k | 64k |
| Baseline | 53.42 | 52.80 | 53.11 | 54.35 | 60.41 | 60.68 | 61.56 | 61.10 |
| + DL | 52.17 | 52.07 | 53.31 | 53.63 | 62.06 | 63.03 | 62.52 | 62.75 |
| + DL & LTA | **54.46** | **55.49** | 54.66 | 55.60 | 63.35 | 64.09 | 64.00 | 63.26 |
| + DL & LTA & ATS | **54.46** | 55.29 | **57.15** | **57.26** | **65.19** | **66.29** | **66.02** | **65.56** |

Hallucination task, specifically the Object-Relation Hallucination subtask, VideoRoPE achieves an impressive 18.0% improvement over existing methods, highlighting its ability to better discern complex spatial interactions. These results underscore VideoRoPE's robustness in solving video hallucination and potential for real-world video analysis.

### 5.5. Ablation Studies

**Ablation Studies on Module Design.**

We conduct ablation experiments on the modules introduced in Section 4, quantitatively evaluating their impact on LongVideoBench and MLVU benchmarks. The experimental results are presented in Tab. 5. The baseline setting, M-RoPE (Wang et al., 2024a), achieves scores of 54.35 on LongVideoBench and 61.10 on MLVU (both using a 64k context length). By progressively integrating the DL (Diagonal Layout), LTA (Low-frequency Temporal Allocation), and ATS (Adjustable Temporal Spacing) modules, our method shows a continuous improvement in performance, achieving enhanced scores of 57.26 on LongVideoBench and 65.56 on MLVU (both using a 64k context length). These results demonstrate the effectiveness of our approach in leveraging spatial-temporal positional information. To refine the analysis of $x$ and $y$ allocation in LTA, we quantitatively evaluate interleaved vs. sequential layouts. We also compare strategies for allocating $t$, $x$, and $y$, including M-RoPE, a uniform interleaved layout, and our VideoRoPE design. Additionally, we explore the optimal ATS scaling factor by varying its value, and further ablate the diagonal layout module to validate the symmetry-based design. See Appendix A.1 for details.

## 6. Conclusion

This paper identifies four key criteria for effective positional encoding: 2D/3D structure, frequency allocation, spatial symmetry, and temporal index scaling. As part of our analysis, through the V-NIAH-D task, we demonstrate that previous RoPE variants are vulnerable to distractors because of lacking proper temporal allocation. As a result, We propose VideoRoPE that uses a 3D structure for spatiotemporal coherence, low-frequency temporal allocation to reduce oscillations, a diagonal layout for spatial symmetry, and adjustable temporal spacing. VideoRoPE outperforms previous RoPE variants in tasks like long video retrieval, video understanding, and video hallucination.

## Acknowledgments

This work was supported by National Key R&D Program of China 2022ZD0161600, Shanghai Artificial Intelligence Laboratory, the Centre for Perceptual and Interactive Intelligence (CPII) Ltd under the Innovation and Technology Commission (ITC)'s InnoHK. Dahua Lin is a PI of CPII under the InnoHK.

## Impact Statement

This paper presents work whose goal is to advance the field of Machine Learning. There are many potential societal consequences of our work, and none of which we feel must be specifically highlighted here.

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

# Appendix

This appendix provides additional resources to further enhance the understanding of our work. In Section A, we present ablation studies and extrapolation experiments extending up to 128k, which are included here due to space constraints in the main text. Section B offers a more detailed discussion of the benchmarks used for evaluation. Section C reviews related work on video LLMs and video haystack retrieval. Section D showcases examples from our proposed **V-NIAH-D** benchmark. In Section E, we provide additional attention visualizations to further support the observations discussed in Figure 3. Finally, Section F provides a more detailed analysis of the frequency allocation and further elaborates on Figure 4.

## A. MORE EXPERIMENTS

### A.1. Supplementary Ablation Experiments

**Ablation Studies on the Scaling Factor $\delta$ for ATS.** We conduct a series of experiments to further investigate the impact of the temporal scaling factor $\delta$ on the alignment between video and text representations. Accurate temporal alignment plays a vital role in enhancing the model's understanding of both semantic and sequential aspects of video-language data. To this end, we evaluate the model's performance across three representative video-language benchmarks—**LongVideoBench**, **MLVU**, and **VideoMME**—by varying the temporal scaling factor $\delta$ from 0.5 to 3.0. As shown in Table 6, we observe a consistent trend across all benchmarks: performance improves as $\delta$ increases, peaking at $\delta = 2$ with an average score of **60.92**. These results suggest that setting $\delta = 2$ strikes the best balance between temporal resolution and semantic alignment, resulting in optimal overall performance.

*Table 6.* Performance under different scaling factors $\delta$ across multiple benchmarks.

| Scaling Factor $\delta$ | LongVideoBench | MLVU | VideoMME | Avg |
|:---:|:---:|:---:|:---:|:---:|
| 0.5 | 50.83 | 59.87 | 58.33 | 56.34 |
| 1.0 | 54.11 | 63.54 | 59.67 | 59.11 |
| 2.0 | 55.50 | 65.59 | 61.67 | **60.92** |
| 3.0 | 53.83 | 63.38 | 60.33 | 59.18 |

**Ablation Studies on x, y Allocation.** To further investigate the impact of different allocation strategies, we conduct quantitative experiments on our proposed **VideoRoPE**, comparing sequential and interleaved allocations of $x$ and $y$. The results, summarized in Table 7, indicate that interleaving $x$ and $y$ leads to superior performance.

We hypothesize that this improvement arises because interleaving maintains the similarity between the $x$ and $y$ dimensions, whereas sequential allocation increases their disparity, thereby hindering model performance.

*Table 7.* Ablation Study on $x$, $y$ Allocation. **VideoRoPE (Sequential)** represents the sequential allocation of $x$ and $y$, following the pattern $x, x, x, \ldots, y, y, y, \ldots$ (similar to M-RoPE (Wang et al., 2024a)). **VideoRoPE (Interleaved)** represents the interleaved allocation, following the pattern $x, y, x, y, \ldots$ (similar to Agrawal et al. (2024)).

| Method | LongVideoBench | | | | MLVU | | | |
|:---:|:---:|:---:|:---:|:---:|:---:|:---:|:---:|:---:|
| | 8k | 16k | 32k | 64k | 8k | 16k | 32k | 64k |
| VideoRoPE(Sequential) | 53.73 | 53.52 | 54.97 | 54.77 | 62.75 | 63.31 | 62.75 | 63.08 |
| VideoRoPE (Interleaved) | **54.46** | **55.29** | **57.15** | **57.26** | **65.19** | **66.29** | **66.02** | **65.56** |

**Ablation Studies on Diagonal Layout Validated on More Benchmarks** To further substantiate our claim that the Diagonal Layout (DL) enhances a model's capability in video understanding tasks, we conduct additional ablation studies on four diverse and challenging benchmarks: **MLVU**, **VideoHallucer**, **V-NIAH**, and **V-NIAH-D**. These benchmarks cover a wide range of evaluation perspectives, from multi-level video question answering to hallucination detection and fine-grained temporal alignment. As shown in Table 8, incorporating the DL module consistently improves performance over the baseline model across all benchmarks. Specifically, we observe notable gains on MLVU and V-NIAH and V-NIAH-D, suggesting that DL effectively facilitates better temporal reasoning and semantic alignment. These results reinforce the generalizability and robustness of the proposed Diagonal Layout design in understanding across various tasks.

*Table 8.* Effect of Diagonal Layout (DL) across multiple benchmarks.

| Method | MLVU | VideoHallucer | V-NIAH | V-NIAH-D |
|--------|------|---------------|--------|----------|
| baseline | 61.56 | 34.3 | 78.67 | 74.67 |
| + DL | **63.03** | **34.8** | **80.44** | **76.44** |

**Ablation Studies on Different Frequency Allocation Strategies**    We compare three different frequency allocation strategies: the *M-RoPE* approach, which emphasizes high-frequency modeling of temporal information and follows a `[t...x...y...]` format; an interleaved and evenly distributed pattern such as `[t t x y x y x y]`; and our proposed *VideoRoPE* method, which prioritizes positional encoding followed by low-frequency temporal modeling, arranged in a `[x y...t...]` format.

We evaluate these approaches on the *LongVideoBench* benchmark under varying context lengths. This benchmark includes a diverse set of video scenarios, ranging from rapidly changing dynamic scenes to slowly evolving static content.

As shown in the results below, our low-frequency temporal allocation consistently outperforms the interleaved `[t t x y x y x y]` pattern on average. This suggests that our frequency design more effectively balances global temporal context modeling with local spatial dynamics, making it better suited to handle a wide variety of video conditions.

*Table 9.* Comparison of different frequency allocation strategies under various context lengths.

| Context | [t...x...y...] | [t t x y x y] | [xy...t...] (Ours) |
|---------|----------------|---------------|--------------------|
| 16k | 60.05 | 59.95 | **62.03** |
| 32k | 59.33 | 58.40 | **59.54** |
| 64k | 58.71 | 57.73 | **59.12** |
| **Avg** | 59.36 | 59.06 | **60.14** |

## A.2. Extrapolation to 128k Experiments

To explore the extrapolation limits of our approach, we extend the visual context during inference to 128k. Specifically, we utilize the **vLLM framework** (Kwon et al., 2023) in Server-API processing mode to enable efficient 128k inference.

Due to the prolonged evaluation time required for 128k processing, we focus on the **LongVideoBench** benchmark. As shown in Table 10, although all four methods exhibit performance degradation at 128k, our proposed **VideoRoPE** experiences the least drop, demonstrating its robustness under extreme extrapolation settings.

*Table 10.* Comparison of model performance at 64k and 128k context lengths for different methods.

| Method | LongVideoBench | |
|--------|------|------|
| | 64k | 128k |
| Vanilla RoPE (Su et al., 2024) | 54.04 | 48.01 |
| TAD-RoPE (Gao et al., 2024) | 53.42 | 45.77 |
| M-RoPE (Wang et al., 2024a) | 54.35 | 51.45 |
| VideoRoPE | **57.26** | **55.64** |

## B. Additional Details on Evaluation Benchmarks

For **long video understanding**, we employ three benchmarks: (1) **LongVideoBench** highlights reasoning questions that depend on long frame sequences, which cannot be effectively addressed by a single frame or a few sparse frames, with durations ranging from 8 seconds to 1 hour. We retain only the questions that are free from subtitles. (2) **MLVU** provides a comprehensive benchmark tailored for assessing the performance of Multimodal Large Language Models in understanding long videos. The dataset features videos lasting between 3 minutes and 2 hours, with nine diverse evaluation tasks. For our analysis, we concentrate on seven multiple-choice tasks, including Topic Reasoning, Anomaly Recognition, Needle QA, Ego Reasoning, Plot QA, Action Order, and Action Count. (3) **Video-MME** stands out as a high-quality benchmark curated for broad scenario coverage, with videos drawn from six key visual domains and 30 subfields. Its dataset spans a wide temporal range, including short clips of 11 seconds and extended videos lasting up to 1 hour.

For **long video retrieval**, we adopt the following two benchmarks: (1) **V-NIAH** is specifically designed to identify highly specific moments within long videos, simulating real-world scenarios where only a small segment of a video is relevant within a vast corpus. The setup follows the same configuration as LongVA, where a "needle" image is inserted at a random position within a "haystack" of 3,000 frames. Each needle image corresponds to a particular question, which is unrelated to the content of the haystack. Each frame is encoded with 144 tokens, and the needle frame is inserted at 0.2 depth intervals. Validation begins at 100 frames, with checks every 200 frames up to 3,000. (2) **Vision Needle-in-a-Haystack with Distractors (V-NIAH-D)**, our proposed method, builds upon V-NIAH by periodically inserting a distractor 200 frames away from the needle. This distractor is semantically similar to the needle, but it remains irrelevant to the specific question being asked. The insertion period for the distractor is calculated using $2 \cdot \pi \cdot 1000000^{32/128} \approx 198.7$. In our experiments, we directly use a period of 200 for distractor insertion. For additional examples, refer to Figure 8.

For the **video hallucination**, we use **VideoHallucer** for evaluation. VideoHallucer classifies hallucinations into two primary types: intrinsic and extrinsic. It further breaks these down into subcategories for detailed analysis, including object-relation, temporal, semantic detail, extrinsic factual, and extrinsic non-factual hallucinations. This framework assesses the model's ability to accurately answer both basic and hallucinated questions about the video content.

# C. More Related Works

**Related Work on Video LLMs (Video Large Language Models)** Video Large Language Models (Video LLMs) build upon the success of image-based vision-language models (VLMs) (Liu et al., 2023a; Zhang et al., 2023b; Dong et al., 2024; Zhang et al., 2024c; Zang et al., 2025; Chen et al., 2023; Chen & Xing, 2024; Liu et al., 2024c;b; Huang et al., 2024; Liu et al., 2024d; Xing et al., 2024; Zhao et al., 2024a; Ding et al., 2025), which align vision and language representations (Radford et al., 2021; Zhang et al., 2024a; Sun et al., 2023) but primarily focus on static images. Extending these models to video requires handling temporal dependencies (Xu et al., 2021; Lei et al., 2021; Bertasius et al., 2021; Huang et al., 2023) and long-form video understanding (Wang et al., 2024c; Chen et al., 2024b; Zhang et al., 2024d). Early video LLMs, such as Wang et al. (2023) and Li et al. (2023), leverage various Video Foundation Models (ViFMs), such as InternVideo (Wang et al., 2022), to extract video attributes, enabling LLM-based question answering. However, their ability to process video content is constrained by the limitations of ViFMs, restricting their effectiveness to short videos. To address this, Luo et al. (2023) introduces a Temporal Modeling Module, allowing end-to-end training of LLMs on video data. Building on this approach, Maaz et al. (2024a) further enhances spatiotemporal modeling to improve video comprehension. Meanwhile, Zhang et al. (2023a) and Lin et al. (2023b) integrate multiple modalities, such as audio and images, to enrich video understanding. These advancements lay the groundwork for processing long videos with greater accuracy. To extend Video LLMs' capabilities to longer content, Lin et al. (2023c) first generates clip-level captions and then employs an LLM to integrate them into a comprehensive video caption, effectively representing the entire video. Various studies, such as those by Li et al. (2024b), Jin et al. (2023), Xu et al. (2024), and Zhang et al. (2025), explore different pooling strategies to reduce the number of video tokens, enabling LLMs to process longer videos more effectively. As the field progresses, there is a growing emphasis on long-form video understanding, exploring techniques such as streaming-based processing (Qian et al., 2025a; Li et al., 2025), memory-augmented models (Qian et al., 2025b; Ding et al., 2024a), and hierarchical representations (Wang et al., 2024e) to efficiently model extended temporal structures for tasks like event-level comprehension (Liu et al., 2024a) and video summarization (Chai et al., 2024).

**Related Work on Video Haystack Retrieval** Originating from the Needle-In-A-Haystack task in Natural Language Processing (Kamradt, 2023; LangChain, 2024), Video haystack tasks aim to locate specific *needle*, the target information, within vast *haystack*, collections of video or multi-modal content (Zhang et al., 2024d; Wang et al., 2024b). In the video domain, VNBench (Zhao et al., 2024b) first introduced a video haystack framework with diverse types of needles, such as subtitles, images, and video clips, specifically designed for retrieval tasks within a three-minute timeframe. V-NIAH (Zhang et al., 2024d) further advanced the field by extending retrieval tasks to long durations of up to one hour, providing tools for comprehensive evaluation. In a broader multi-modal domain, MMNeedle (Wang et al., 2025) modeled the retrieval task as locating the exact coordinates of a sub-image within a larger multi-image haystack, while MM-NIAH (Wang et al., 2024b) introduced a setting where both haystack and needle could include images or text, emphasizing interleaved retrieval capabilities.

However, video haystack retrieval still lags in difficulty compared with the QA or retrieval task in NLP (Hsieh et al., 2024; Yuan et al., 2024). Few attempts exist to enhance the discriminability of evaluations, such as multi-NIAH (LangChain, 2024; Hsieh et al., 2024) or NIAH with distractors (Hsieh et al., 2024). Although VHs (Wu et al., 2024b) introduces distractors into

the multi-image haystack setting, where needles and distractors are randomly inserted, VHs still did not consider temporal dependencies between video frames or more structured approaches to task evaluation. Our work builds upon V-NIAH by introducing distractors in a systematic, periodic manner based on rotary bases.

## D. V-NIAH-D Examples

Figure 8 illustrates the five VQA needles used in V-NIAH-D, along with their corresponding distractors. The visual questions and their respective answers are the only components in V-NIAH-D that require human annotation, making it an ideal benchmark for evaluating the long-context reasoning capabilities of LMMs.

## E. Supplementary Attention Analysis

To further explain the attention pattern in Figure 3, we present additional visual analysis in Figure 9. An attention analysis comparing M-RoPE and VideoRoPE is conducted using 8k-context input, with video tokens from the same frame aggregated through average pooling. As a result, one tick on the axis represents a single frame during inference. The evaluation setup for Figure 3 is the same as for Figure 9. M-RoPE relies on high-frequency temporal modeling, limiting it to local information and hindering effective needle identification for question answering. On the other hand, VideoRoPE employs low-frequency temporal modeling, allowing it to capture long-range dependencies and successfully identify the needle for accurate responses.

## F. Supplementary Explanation on Frequency Allocation

This section provides a detailed explanation of the supplementary information related to Figure 4, highlighting the advantages of our frequency allocation. Consider a RoPE-based LLM with a head dimension size of 128, corresponding to 64 rotary angles $\theta_n$ across different dimensions. In each illustration, we visually represent the function $\cos(\theta_n t)$ for 3 dimensions using parallel blue planes.

**(a)** For M-RoPE (Wang et al., 2024a), temporal dependency is modeled using the first 16 rotary angles, which exhibit higher frequency and greater oscillation. Taking the last 3 rotary angles as an example, the position embedding for temporal modeling undergoes significant distortion due to periodic oscillations (Men et al., 2024), as these dimensions have shorter monotonous intervals. Lower dimensions have even shorter intervals. Notably, because the oscillation is periodic, two distant positions can have nearly identical position embeddings, resembling a hash collision, as shown by the red planes. This phenomenon is why distractors can easily mislead the model.

**(b)** In contrast, for VideoRoPE, temporal dependency is modeled using the last 16 rotary angles, which have much wider monotonous intervals. Taking the first 3 rotary angles as an example, the position embedding for temporal modeling is free ferom oscillation (Men et al., 2024). As a result, the misleading effects of distractors are significantly suppressed.

## V-NIAH-D Needles and Distractors

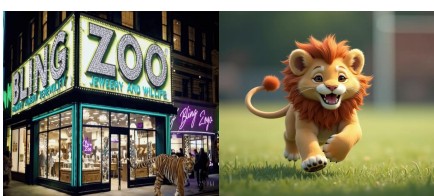

Question: Find the frame with the word 'zoo'. What is the animal outside the zoo shop?
A. lion
B. tiger
C. horse
D. dog
Answer with the option's letter from the given choices directly.

Answer: B

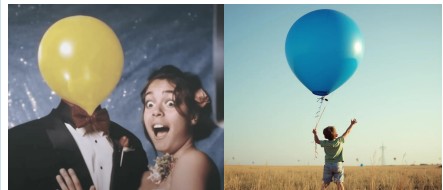

Question: Find the frame of a couple in a wedding... What is the color of that ballon?
A. Yellow
B. Red
C. Blue
D. White
Answer with the option's letter from the given choices directly.

Answer: A

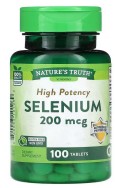 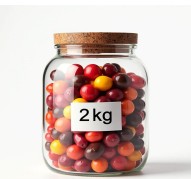

Question: Find the frame with the image of Selenium tablets. How many mg does each tablet contain?
Answer the question using a single word or phrase.

Answer: 200

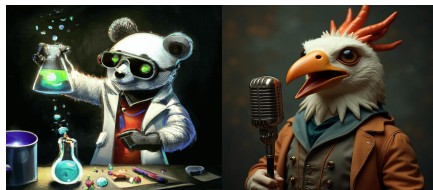

Question: Find the frame of a scientist. The scientist is a...
A. Bird
B. Elephant
C. Panda
D. Dog
Answer with the option's letter from the given choices directly.

Answer: C

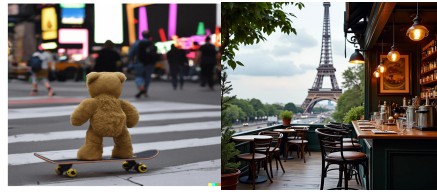

Question: Find the frame of a teddy bear. Where is this teddy bear?
A. Times Square
B. Eiffel Tower
C. Taj Mahal
D. Sydney Opera House
Answer with the option's letter from the given choices directly.

Answer:  A

*Figure 8.* Five visual question-answering problems along with their corresponding needle and distractor.

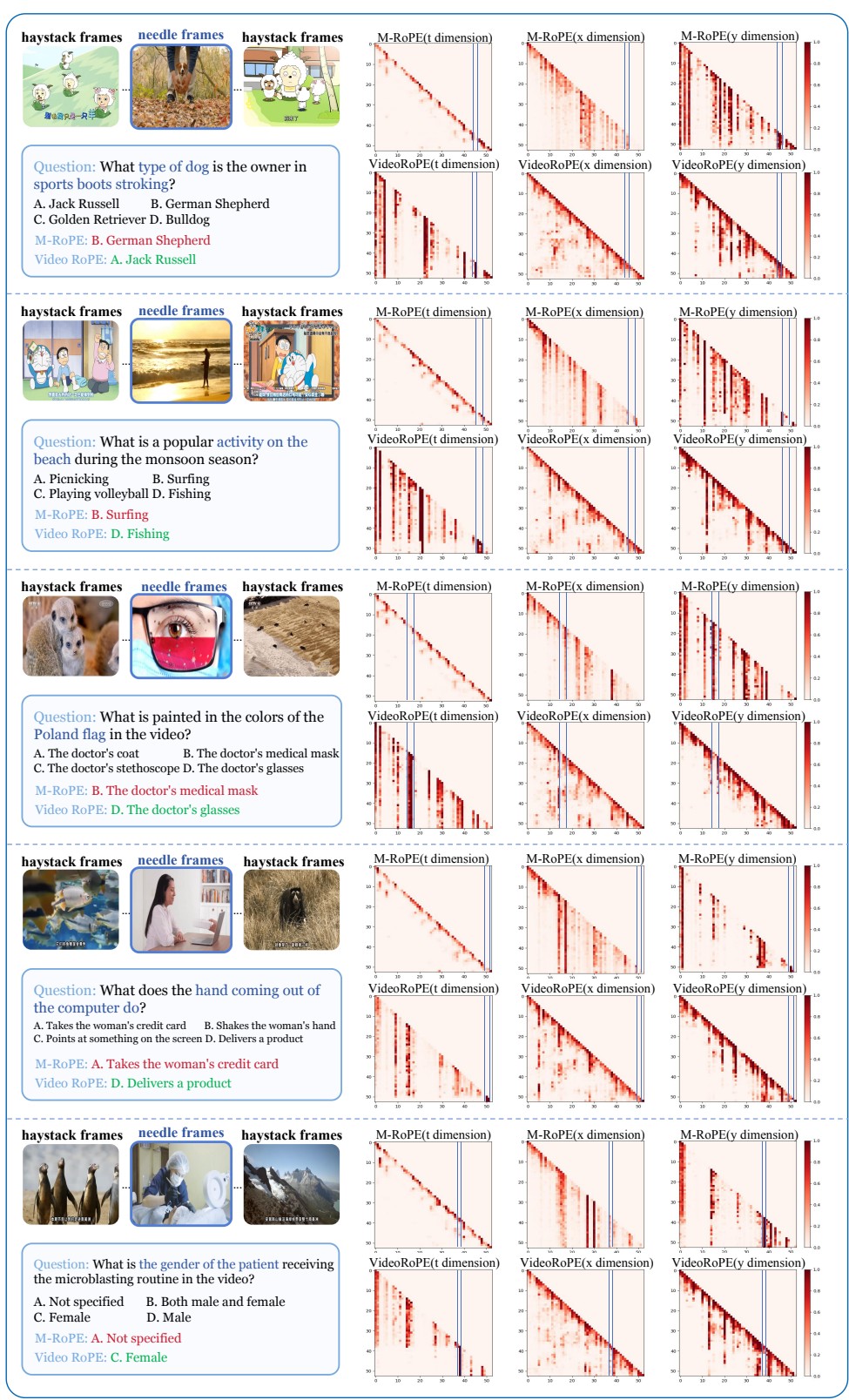

*Figure 9.* Additional visual analysis of attention.

