# OpenReview forum: "VideoRoPE: What Makes for Good Video Rotary Position Embedding?"
_ICML.cc/2025/Conference — ICML 2025 oral_

### Official Review · Reviewer_3qrY · 2025-02-20

**Overall Recommendation:** 5

**Summary:**

In this work, the authors study criteria for video position embedding and propose a rotary position embedding method. They claim that a good video position embedding should handle 3D structure, and appropriate frequency allocation to prevent embedding collision, spatial symmetry and temporal index scaling between text and visual tokens. According to these criteria, they build their method upon M-RoPE (Wang et al., 2024b). They propose low-frequency temporal allocation to avoid embedding collision, and diagonal position ID layout to make the token distances more balanced, and adjustable temporal scaling that controls temporal interval between visual tokens with a hyperparameter. They validate the proposed idea on public benchmarks: long video understanding, long video retrieval, and video hallucination.



## update after rebuttal
My major concerns are all resolved by the rebuttal. I have increased my rating to strong accept.

**Claims And Evidence:**

Some of the claims are not fully supported by clear and convincing evidence.
- L266-270, second column: “This periodicity creates “hash collisions” (red planes), where distant positions share near-identical embeddings, making the model susceptible to distractor influence.” $\rightarrow$ Why such collisions make the model susceptible to distractor influence? I understand the proposed method helps avoiding such collisions and improves V-NIAH-D performance significantly upon M-RoPE baseline. However, I do not see any logical connection between the collision and distractor influence. Can the authors provide either logical explanations or more direct empirical evidence on this?
- L413-414, first column: “Fig. 7 (a) and (b) demonstrate that the proposed V-NIAH-D is more challenging than V-NIAH.” $\rightarrow$ It is quite arguable that V-NIAH-D is more challenging that V-NIAH by merely looking into Fig. 7.  Can the authors provide quantitative measure of challengingness of these two datasets?

**Essential References Not Discussed:**

I do not see any missing essential references not discussed.

**Experimental Designs Or Analyses:**

I have checked all experimental designs and analysis. They all seem to be valid.

**Methods And Evaluation Criteria:**

The proposed method and evaluation criteria make sense for the problem of video positional embedding.

**Other Comments Or Suggestions:**

I list a few errors, typos, and awkward sentences below.
- L214-216, second column: “M-RoPE (Wang et al., 2024b) uses dimensions to model temporal, horizontal, and vertical dimensions sequentially” $\rightarrow$ “uses dimensions” seems a bit awkward.
- L271-273, second column: “The visualized relationship between the periodicity, monotonicity, and temporal modeling.” $\rightarrow$ Not an English sentence.
- L357, first column: “…different modes that ….” $\rightarrow$ models

**Other Strengths And Weaknesses:**

- Strengths: This paper is well-motivated and the proposed method is sensible. The experimental results demonstrate the clear merit of the proposed video position embedding method. The paper is well-organized and easy to follow in general.
- Weaknesses: There is missing evidence for some of the claims made. Please address the “Claims and Evidence” section. Although the paper is easy to follow in general, there are some errors, typos, and awkward sentences. Furthermore, there are some unclear points in the paper. The presentation quality could be improved to a more professional level.

**Questions For Authors:**

There are some unclear points in the paper. I list a few below.
- In Fig. 2 right side, what do the axes mean and each cell color mean?
- In Fig. 3 of the main paper, and Fig. 9 of the supplementary material, How do we interpret the results? For instance, in L165-176, second column, what does it mean by “the needle is located primarily through vertical positional information”? I do not understand this paragraph. Can the authors elaborate on this paragraph?
- In L298-303, first column, what does it mean by “creating a stack in the bottom-left corner”? I do not understand this paragraph as well. Can the authors elaborate on this paragraph?
- Can the authors give more intuitive explanation on Fig. 6?

**Relation To Broader Scientific Literature:**

The key contribution of this paper is related to the general multi-modal LLM field. The findings and proposed method could be helpful for the general audience of the field.

**Theoretical Claims:**

There is no theoretical claim made.

---

> ### Author Rebuttal · Authors · 2025-04-01
>
> Dear Reviewer 3qrY,
>
> Thanks for your valuable feedback. We sincerely thank all reviewers for acknowledging that this paper is novel (t9dP, juMQ),  well-motivated (t9dP, 26sL, 3qrY), significant improvements (t9dP, 26sL, 3qrY), well-written (t9dP, juMQ, 26sL, 3qrY), and easy to follow (t9dP, 3qrY). Below we address your questions:
>
> > Q1: L266-270, second column. Why such collisions make the model susceptible to distractor influence?...Can the authors provide either logical explanations or more direct empirical evidence on this?
>
> We appreciate your request for clarification. As shown in Fig. 4 and Appendix F, we have linked position embedding collisions in M-RoPE to distractor influence. Due to high-frequency oscillations, distant positions can share nearly identical embeddings (red planes in Fig. 4), making the model vulnerable to distractors. In contrast, VideoRoPE avoids such oscillations, enabling more robust temporal modeling. We have also conducted an ablation study (Table 5), confirming that LTA(Low-frequency Temporal Allocation)—by preventing temporal embedding collisions—improves video understanding performance.
>
> > Q2: L413-414, first column:  It is quite arguable that V-NIAH-D is more challenging than V-NIAH by merely looking into Fig. 7. Can the authors provide quantitative measure of challengingness of these two datasets?
>
> In the table below, we provide the quantitative performance gaps observed in various models, including our Video-RoPE, tested on both V-NIAH and V-NIAH-D datasets:
>
> |Model|V-NIAH|V-NIAH-D|
> |-|-|-|
> |LLaVA-NeXT-Video-7B|14.66|3.99|
> |LongVA|90.88|80.44|
> |Qwen2.5-VL-7B-VideoRoPE|86.44|86.22|
>
> These results indicate a significant performance drop on V-NIAH-D for most methods (-10.67 for LLaVA-NeXT-Video-7B and -10.44 for LongVA), supporting our claim that V-NIAH-D is more challenging than V-NIAH. Also, our VideoRoPE effectively mitigates the misleading influence of distractors in V-NIAH-D (-0.22 merely). We will include this quantitative comparison in our revised manuscript.
>
> > Q3:  I list a few errors, typos, and awkward sentences below.
>
> Thank you for highlighting language issues and awkward expressions. We will carefully revise the manuscript to correct these:
> - "uses dimensions" (L214-216) will be corrected to "uses different dimensions".
> - "The visualized relationship between the periodicity, monotonicity, and temporal modeling" (L271-273) will be corrected to "The relationship between periodicity, monotonicity, and temporal modeling is visualized in Fig 4."
> - "modes" will be corrected to "models" (L357).
>
> > Q4: In Fig. 2 right side, what do the axes mean and each cell color mean?
>
> Thank you for pointing that out. Fig. 2 and the last row of Fig. 7 are indeed the same, both showing M-RoPE and VideoRoPE performance on V-NIAH and V-NIAH-D benchmarks. The x-axis indicates context length, the y-axis shows frame depth, and the green-to-red color bar reflects needle retrieval accuracy—from perfect to zero.
>
> > Q5: In Fig. 3 of the main paper, and Fig. 9 of the supplementary material, How do we interpret the results? For instance, in L165-176, second column, what does it mean by “the needle is located primarily through vertical positional information”? I do not understand this paragraph. Can the authors elaborate on this paragraph?
>
> In Fig 3, we can derive the fact from different component of the attention score, that M-RoPE locates and retrieves the needle using the component with respect to the feature dimension capturing vertical positional information, namely the product of the last 48 dimension of Q and K in self-attention. This goes against the design intention of M-RoPE.
>
> > Q6: In L298-303, first column, what does it mean by “creating a stack in the bottom-left corner”? I do not understand this paragraph as well. Can the authors elaborate on this paragraph? Can the authors give more intuitive explanation on Fig. 6?
>
> The three subfigures in Figure 6 represent the schema of Vanilla RoPE, M-RoPE, and VideoRoPE when encoding text-video-text input. The three dimensions of each subfigure represent the sequential indices (both textual and temporal indices), and the horizontal and vertical positions in each frame. We find that each dimension of Vanilla RoPE increases with the input token indices, regardless of the existence and the spatiotemporal characteristics of video, which is a straight diagonal line in 3D space. After that, M-RoPE uses part of dimensions to represent the spatial information of the video, but the horizontal and vertical position indices do not change with the video frames. Therefore, it appears as a vertical stack in the bottom-left corner in 3D space, deviating from the diagonal direction, different from the pre-trained features in the text. Comparatively, VideoRoPE allows the horizontal and vertical position indices of each frame to change with the frame indices, while maintaining the diagonal features, which makes it easier to transfer LLM to the video mode.

---

> > ### Comment · Reviewer_3qrY · 2025-04-07
> >
> > I have read the rebuttal and other reviews. I appreciate the rebuttal. Most of my concerns are resolved by the rebuttal. However, I am still confused about the "distractor influence". How do the authors define a distractor?

---

> > > ### Author Response · Authors · 2025-04-07
> > >
> > > Dear Reviewer 3qrY,
> > >
> > > Thank you for your feedback! We are glad that most of your concerns have been addressed.  As for your last question regarding the definition and rationale behind distractors in our proposed benchmark, **V-NIAH-D**. We provide a detailed clarification below.
> > >
> > > In **V-NIAH-D (Visual Needle-in-a-Haystack with Distractors)**, a *distractor* is specifically defined as a frame that is both **visually and semantically similar to the ground-truth needle** yet remains **irrelevant to the query posed**. These distractors are intentionally introduced to rigorously evaluate the model's ability in accurate interpretation and robust long-context reasoning, by providing challenging *hard negatives* that closely mimic the correct frame but do not answer the query.
> > >
> > > Distractors are carefully selected to share strong visual and contextual similarity with the needle, such as similar object categories, scene structures, or visual themes. Despite this resemblance, they are irrelevant to the specific question being asked. This ensures that the correct answer remains unambiguous, while also rendering superficial pattern matching or short-term heuristics ineffective.
> > >
> > > To further increase the difficulty, distractors are inserted every 200 frames—a value guided by the analytical approximation:
> > >
> > > $2 \cdot \pi \cdot 1000000^{32/128} \approx 198.7$
> > >
> > > This periodic strategy ensures that distractors and the needle share similar positional embeddings, thus introducing interference at the position-encoding level during queries.
> > >
> > > We source these distractors from *Google Image Search* and *Flux*, yielding visually rich and naturally diverse content that is plausibly confusable with the needle, yet explicitly filtered to maintain irrelevance to the query.
> > >
> > > This design significantly reduces the likelihood of random guessing and encourages models to engage in genuine long-context reasoning, rather than relying on superficial heuristics or shortcut cues.
> > >
> > > We have illustrated and discussed these concepts clearly in **Figure 8** and **Appendix B** of the manuscript. We welcome any further concerns you might have. In light of our response, we would greatly appreciate your review and a potential adjustment to your score. Thank you again for your time and thoughtful feedback.

---

### Official Review · Reviewer_26sL · 2025-03-12

**Overall Recommendation:** 4

**Summary:**

This paper identifies four key factors in extending position encodings from images to videos: spatio-temporal structures, frequency allocation, spatial symmetry, and temporal indexing. Drawing from these observations, the authors propose VideoRoPE, which (1) uses low-frequency temporal allocation to mitigate periodic oscillations, (2) applies a diagonal layout to maintain spatial symmetry, and (3) adopts adjustable temporal spacing to decouple temporal from spatial indexing. The method shows consistent performance improvements in long video retrieval, video understanding, and video hallucination tasks.

**Claims And Evidence:**

The four key aspects conceptually sound but stronger empirical validation is required, especially on frequency allocation and spatial symmetry.

Temporal allocation: The authors’ figures and retrieval results (e.g., Fig. 6) support that low-frequency allocation reduces periodic oscillations, but explicitly showing some specific failure cases of high-frequency allocation would strengthen the claim.

Spatial symmetry: The only reference for spatial symmetry (Su et al., 2024b) is very weak. It’s just a blog not a peer-reviewed source. I understand the concept of the need of spatial symmetry but I think the authors should have provided more empirical qualitative study of how spatial inconsistency becomes problematic.

**Essential References Not Discussed:**

[a] Heo et al., “Rotary Position Embedding for Vision Transformer,” ECCV, 2024.

This reference would provide additional context for 2D extensions of RoPE in vision tasks.

**Experimental Designs Or Analyses:**

Most of the experiments on different video tasks appear valid and sufficiently controlled, except scaling factor ablation in Tab.6.
The performace seems to be sensitive to the scaling factor $\delta$ on LongVideoBench. Does the optimal scaling factor \delta vary significantly across benchmarks, or is it stable? Did you tune it separately for each dataset?

**Methods And Evaluation Criteria:**

The authors’ approach—incorporating low-frequency temporal allocation, a diagonal layout, and adjustable temporal spacing—is broadly reasonable. However, one concern remains regarding low-frequency temporal allocation.

While using only low frequencies may help capture global or long-range context, it potentially risks missing short-term or local dynamics, which are often critical in videos displaying rapid changes (e.g., fast-moving objects or abrupt scene transitions).

A related question is why the temporal encoding frequencies are not interleaved with the spatial frequencies (like the 2D RoPE approach), balancing both low and high frequencies for the time dimension. For example, one could allocate 32, 48, 48 frequencies for (t,x,y) and interleave them as [t t x y x y x y … t t x y x y x y] or [t x y t x y x y … t x y t x y x y] to preserve the ability to capture both long- and short-term variations.

**Other Comments Or Suggestions:**

- Captions in Fig.2 should be more informative. Especially, on right figure, the caption should explain what the x, y axes, and color indicates

- In Eq.7, I think the equation needs to be correct as, $(t,x,y)=(\tau + (\delta - 1)T_v, \tau + (\delta - 1)T_v, \tau + (\delta - 1)T_v)$ if $T_s+T_v \leq \tau \leq T_s+T_v+T_e$

**Other Strengths And Weaknesses:**

Strengths:
- The paper is well-written, motivating the problem clearly.
- Empirical results consistently demonstrate improvements across diverse tasks.

Weaknesses:
- Please see above.

**Questions For Authors:**

Please see above.

**Relation To Broader Scientific Literature:**

The core findings and solutions (i.e., diagonal layout) for spatial symmetry was already proposed in (Su et al, 2024b). Personally, I think the proposed method essentially extends RoPE-Tie-V2(Su et al, 2024b) to the temporal domain. The newly featured low-frequency temporal allocation and temporal spacing are technically less significant or already proposed in (Li et al., 2024), respectively.

**Theoretical Claims:**

Theoretical claims appear to sound although further empirical and qualitative analysis should provide compelling support.

---

> ### Author Rebuttal · Authors · 2025-04-01
>
> Dear Reviewer 26sL,
>
> Thanks for your valuable feedback. We sincerely thank all reviewers for acknowledging that this paper is novel (t9dP, juMQ),  well-motivated (t9dP, 26sL, 3qrY), significant improvements (t9dP, 26sL, 3qrY), well-written (t9dP, juMQ, 26sL, 3qrY), and easy to follow (t9dP, 3qrY). Below we address your questions:
>
> > Q1: Temporal allocation...showing some specific failure cases of high-frequency allocation...
>
> Failure cases of high-frequency allocation in M-RoPE have shown in Fig. 3 and Appendix E. Specifically, the M-RoPE responses illustrate its limitations, with attention visualizations revealing that high-frequency encoding hinders long-range temporal dependencies, favoring local patterns instead.
>
> > Q2: Spatial symmetry...more empirical qualitative study of how spatial inconsistency becomes problematic
>
> We have quantitatively shown the impact of spatial symmetry in Tab. 5 (second row), with a +1.65 gain on MLVU under 64k context over M-RoPE. To further validate its importance, we evaluated it on 4 benchmarks with consistent improvements supporting our claim.
>
> |Method|MLVU|VideoHallucer|V-NIAH|V-NIAH-D|
> |-|-|-|-|-|
> |baseline|61.56|34.3|78.67|74.67|
> |+ DL|63.03|34.8|80.44|76.44|
>
> > Q3: While using only low frequencies...potentially risks missing short-term or local dynamics...
>
> See Q4.
>
> > Q4: Why the temporal encoding frequencies are not interleaved with the spatial frequencies ... For example, ... [t t x y x y x y … t t x y x y x y] or [t x y t x y x y … t x y t x y x y] to preserve the ability to capture both long- and short-term variations.
>
> We trained a model using the "[t t x y x y x y]" format and conducted additional comparative experiments with varying context lengths on the LongVideoBench benchmark, which features a wide range of video scenarios—including both rapidly changing dynamic content and slowly evolving scenes. Our results show below that, on average, low-frequency temporal allocation consistently outperforms the "[t t x y x y x y]" arrangement. This suggests that our frequency design effectively balances the modeling of both global context and local dynamics across diverse video conditions. As for the "[t x y t x y x y … t x y t x y x y]" pattern, we plan to report on it in future updates.
>
> |Context Length|[t...x...y...]|[t t x y x y x y]|[xy...t...] (Ours)|
> |-|-|-|-|
> |16k|60.05|59.95|62.03|
> |32k|59.33|58.40|59.54|
> |64k|58.71|57.73|59.12|
> |Avg|59.36|59.06|60.14|
>
> > Q5: ...The performance seems to be sensitive to the δ on LongVideoBench. Does the optimal δ vary significantly across benchmarks, or is it stable? Did you tune it separately for each dataset?
>
> Table 6 presents the ablation study of the scaling factor on LongVideoBench. Additionally, we report the average results on other benchmarks, MLVU and VideoMME, in the table below. Our analysis indicates that the optimal δ remains consistent across various tasks. We would also like to clarify explicitly in the revised manuscript that we did not tune this δ separately for each individual dataset.
>
> |δ|LongVideoBench|MLVU|VideoMME|Avg|
> |-|-|-|-|-|
> |0.5|50.83|59.87|58.33|56.34|
> |1|54.11|63.54|59.67|59.11|
> |2|55.50|65.59|61.67|**60.92**|
> |3|53.83|63.38|60.33|59.18|
>
> > Q6-1: The core findings and solutions (i.e., diagonal layout) for spatial symmetry was already proposed in (Su et al, 2024b). Personally, I think the proposed method essentially extends RoPE-Tie-V2(Su et al, 2024b) to the temporal domain.
>
> This is not true. RoPE-Tie-V2 (Su et al, 2024b) introduced the concept of spatial symmetry but did not provide experimental validation, nor did it discuss the diagonal layout. Additionally, RoPE-Tie-V2 cannot handle streaming video inputs (as discussed by reviewer juMQ regarding streaming scenarios)
>
> > Q6-2: The newly featured low-frequency temporal allocation and temporal spacing are technically less significant or already proposed in (Li et al., 2024), respectively.
>
> This is not true. Li et al. (2024) primarily focuses on enhancing video models from a data perspective, using only temporal reasoning data in pure text. Their work does not address low-frequency temporal allocation or temporal spacing—both of which are novel contributions introduced by our method. We will clearly highlight these distinctions in our revised manuscript.
>
> > Q7: [a] Heo et al., should be cite.
>
> We will cite this work in our related work section. This paper is designed for 2D image tasks. On the contrary, our VideoRoPE is designed for 3D video tasks.
>
> > Q8: Captions in Fig.2
>
> Thank you for pointing that out. Fig. 2 and the last row of Fig. 7 are indeed the same, both showing M-RoPE and VideoRoPE performance on V-NIAH and V-NIAH-D benchmarks. The x-axis indicates context length, the y-axis shows frame depth, and the green-to-red color bar reflects needle retrieval accuracy—from perfect to zero.
>
> > Q9: Typo in Eq.7
>
> Thank you for identifying this typo. We will correct and clarify this equation in our revised submission.

---

> > ### Comment · Reviewer_26sL · 2025-04-08
> >
> > Thank you for your detailed response. After going through the rebuttal, I found that all my concerns are resolved. I strongly recommend adding new results, e.g., streaming videos, temporal allocation, $\delta$ ablation, etc., to the final manuscript. I raise my rating to accept.

---

> > > ### Author Response · Authors · 2025-04-08
> > >
> > > Dear Reviewer 26sL,
> > >
> > > Thanks for increasing the score, and we appreciate it. We are encouraged by your recognition of the quality and significance of our work. Following your recommendation, we will include the new results (e.g., results on streaming video benchmarks, ablation studies on temporal allocation and $\delta$, etc) in our final version.
> > >
> > > Regards,
> > >
> > > Authors

---

### Official Review · Reviewer_juMQ · 2025-03-13

**Overall Recommendation:** 4

**Summary:**

VideoRoPE is a position embedding method designed for video large language models. It addresses four key issues: 3D structure, frequency allocation, spatial symmetry, and temporal index scaling. The authors demonstrate through a new benchmark V-NIAH-D that existing methods perform poorly when distractors are present.
VideoRoPE solves these problems through low-frequency temporal allocation, diagonal layout, and adjustable temporal spacing, outperforming other RoPE variants in video retrieval, understanding, and hallucination benchmarks.

**Claims And Evidence:**

Yes. VideoRoPE's superior performance is demonstrated through comprehensive evaluations across multiple benchmarks with significant improvements (+12.44% on V-NIAH-D, +11.9% on VideoHallucer).

**Essential References Not Discussed:**

The paper covers most of the essential references related to position embeddings for transformer models and video understanding. However, there are a few relevant works that are not cited or discussed that could provide additional context:

[1] Chai, W., Song, E., Du, Y., Meng, C., Madhavan, V., Bar-Tal, O., ... & Manning, C. D. (2024). Auroracap: Efficient, performant video detailed captioning and a new benchmark. arXiv preprint arXiv:2410.03051.

[2] Li, K., Wang, Y., He, Y., Li, Y., Wang, Y., Liu, Y., ... & Qiao, Y. (2024). Mvbench: A comprehensive multi-modal video understanding benchmark. In Proceedings of the IEEE/CVF Conference on Computer Vision and Pattern Recognition (pp. 22195-22206).

[3] Tang, Y., Guo, J., Hua, H., Liang, S., Feng, M., Li, X., ... & Xu, C. (2024). VidComposition: Can MLLMs Analyze Compositions in Compiled Videos?. arXiv preprint arXiv:2411.10979.

**Experimental Designs Or Analyses:**

Yes, I did. The V-NIAH-D benchmark with periodic distractors effectively tests robustness to frequency-based issues. And the evaluations are comprehensive, testing across multiple datasets and context lengths (8k-64k) to assess both in-distribution and extrapolation capabilities.

**Methods And Evaluation Criteria:**

Yes. The proposed method directly addresses video position embedding challenges, the new V-NIAH-D benchmark effectively tests model robustness to distractors, and the evaluation uses appropriate benchmarks covering understanding, retrieval, and hallucination tasks across multiple context lengths, which make sense to me.

**Other Comments Or Suggestions:**

The paper could potentially be enhanced by discussing additional benchmarks that, while not included in the current comparison, might provide valuable insights, such as those works referenced in the "Essential References Not Discussed" section.

**Other Strengths And Weaknesses:**

Other Strengths:
S1: VideoRoPE addresses key issues of current position embedding for video-LLM including 3D structure, frequency allocation, spatial symmetry, and temporal index scaling.
S2: The paper has very strong visualizations to explain frequency allocation and position embedding.
S3: The V-NIAH-D benchmark is a valuable contribution that provides a way to test model robustness to distractors.

Other weakness: no obvious drawbacks.

**Questions For Authors:**

Q1. How would VideoRoPE need to be adapted for streaming video understanding tasks where the full temporal context isn't available upfront?
Since there are many new datasets and benchmarks [1, 2] for streaming video understanding, which is more and more important.

[1] Yang, Z., Hu, Y., Du, Z., Xue, D., Qian, S., Wu, J., ... & Xu, C. (2025). SVBench: A Benchmark with Temporal Multi-Turn Dialogues for Streaming Video Understanding. arXiv preprint arXiv:2502.10810.

[2] Lin, J., Fang, Z., Chen, C., Wan, Z., Luo, F., Li, P., ... & Sun, M. (2024). Streamingbench: Assessing the gap for mllms to achieve streaming video understanding. arXiv preprint arXiv:2411.03628.

**Relation To Broader Scientific Literature:**

VideoRoPE extends position embedding research from text-only models to video, addressing the unique challenge of handling both spatial and temporal dimensions in one attention space. It builds on long-context understanding work (like LongRoPE) and attention mechanism research (particularly attention sinks), while making novel contributions to frequency allocation for effectively modeling video's complex spatio-temporal structure.

**Theoretical Claims:**

Yes, I did. This paper doesn't contain formal mathematical proofs or theoretical claims requiring verification. These equations correctly describe implementation details of vanilla RoPE, M-RoPE, and VideoRoPE, but they serve as explanations rather than theoretical proofs.

---

> ### Author Rebuttal · Authors · 2025-04-01
>
> Dear Reviewer juMQ,
>
> Thanks for your valuable feedback. We sincerely thank all reviewers for acknowledging that this paper is novel (t9dP, juMQ),  well-motivated (t9dP, 26sL, 3qrY), significant improvements (t9dP, 26sL, 3qrY), well-written (t9dP, juMQ, 26sL, 3qrY), and easy to follow (t9dP, 3qrY). Below we address your questions:
>
> > Q1: The paper covers most of the essential references related to position embeddings for transformer models and video understanding. However, there are a few relevant works that are not cited or discussed that could provide additional context.
>
> Thank you for the suggestion. We will incorporate citations and a discussion of Auroracap, MVBench, and VidComposition into the revised manuscript. These benchmarks, which evaluate video large language models, offer valuable insights into detailed video captioning (Auroracap), general video understanding (MVBench), and compositional structure analysis (VidComposition). We also report the performance of our VideoRoPE on these benchmarks, see Q2 below.
>
> > Q2:  The paper could potentially be enhanced by discussing additional benchmarks that, while not included in the current comparison, might provide valuable insights, such as those works referenced in the "Essential References Not Discussed" section.
>
> Thanks for your suggestion. We evaluate our method on the Video-Detailed-Caption (VDC) benchmark proposed by Auroracap and MVBench. Since VidComposition has not yet been open-sourced, we will evaluate it once the benchmark is released.
>
> |Model|VDC|MVBench|
> |-|-|-|
> |Auroracap-7B|38.2|-|
> |Vanilla RoPE|43.0|67.1|
> |TAD-RoPE|43.8|66.9|
> |M-RoPE|**44.0**|67.8|
> |VideoRoPE|**44.0**|**68.4**|
>
> > Q3: How would VideoRoPE need to be adapted for streaming video understanding tasks where the full temporal context isn't available upfront? Since there are many new datasets and benchmarks [1, 2] for streaming video understanding, which is more and more important.
>
> Thank you for this insightful question regarding the adaptation of VideoRoPE to streaming video understanding tasks. We acknowledge the growing importance of streaming video understanding, as it closely aligns with how humans naturally perceive the world. Accordingly, we will reference the works you mentioned in our manuscript and discuss the significance of streaming modeling for video understanding tasks.
>
> By design, VideoRoPE effectively supports streaming input scenarios, in contrast to RoPE-TIE-V2, which requires prior knowledge of the video's total length and thus cannot accommodate streaming data. Although VideoRoPE was not explicitly trained on streaming video data, we evaluated its effectiveness for streaming modeling on the StreamingBench Benchmark[2]. Specifically, we focused on the Real-Time Visual Understanding (RTVU) setting relevant to streaming modeling, as presented in the table below. The results demonstrate that VideoRoPE still exhibits superior performance in streaming video scenarios.
>
> |Model|StreamingBench (RTVU)|
> |-|-|
> | Vanilla RoPE|75.0|
> | TAD-RoPE|75.8|
> | M-RoPE|76.2|
> | VideoRoPE|**77.1**|

---

> > ### Comment · Reviewer_juMQ · 2025-04-09
> >
> > Thanks for your comprehensive rebuttal. It shows the effectiveness of VideoRoPE across broader benchmarks and its potential in streaming tasks. My concerns have been addressed.

---

> > > ### Author Response · Authors · 2025-04-09
> > >
> > > Dear Reviewer juMQ,
> > >
> > > We sincerely appreciate your valuable feedback, which has helped us clarify and strengthen our work. We are glad that we have addressed your concerns and will incorporate your points into our final version.
> > >
> > > Regards,
> > >
> > > Authors

---

### Official Review · Reviewer_t9dP · 2025-03-16

**Overall Recommendation:** 4

**Summary:**

This paper proposes VideoRoPE, a new positional embedding for videos. VideoRoPE extends the 1D RoPE to 3D cases to encode spatiotemporal information. M-RoPE used in QWen2-VL (Wang et al., 2024b) employ 3D structure, dividing the feature dimensions into distinct subsets for spatiotemporal encoding. However, the authors argue that the allocation of feature dimensions is suboptimal. To address this issue, they propose to assign higher dimensions (lower frequencies) to the temporal axis, enhancing long-range temporal dependence modeling without interference from oscillations. Extensive experiments have been conducted to compare VideoRoPE against RoPE variants such as Vanilla RoPE, TAD-RoPE, and M-RoPE. The results on LongVideoBench and MLVU demonstrate VideoRoPE’s superior performance in long video understanding, retrieval, and hallucination.

**Claims And Evidence:**

The paper makes a strong claim that their proposed VideoRoPE method is superior for video understanding tasks compared to existing RoPE variants. The claims are generally well-supported by the evidence presented. For example, VideoRoPE consistently outperforms Vanilla RoPE, TAD-RoPE, and M-RoPE across multiple popular video understanding benchmarks and the proposed Visual Needle-In-AHastack-Distractor benchmark.

**Essential References Not Discussed:**

No

**Experimental Designs Or Analyses:**

The experimental design looks sound. The experiments are quite extensive in terms of the number of benchmark and tasks evaluated. The main experiments include results on Long Video Understanding (covering 3 standard video QA benchmarks), results on Long Video Retrieval (the needle test), results on Video Hallucination. Moreover, ablation studies are provided to further justify the design choices.

**Methods And Evaluation Criteria:**

Method:

The key idea of VideoRoPE is to improve the frequency allocation of the existing M-RoPE. Allocating higher dimensions (lower frequencies) to the temporal axis is a well-motivated strategy. It aims to prioritize temporal modeling and capture long-range dependencies, which are essential for video understanding. The justification based on the "global branch" corresponding to higher dimensions from other works also lends credibility to this. The diagonal layout is intended to maintain spatial symmetry and prevent bias toward input order, which is  reasonable too as it ensures that visual input receives equal contextual influence from preceding and subsequent textual information.

Evaluation Criteria:

The paper evaluates VideoRoPE on a diverse set of video understanding benchmarks including Video-MME, MLVU, etc. The paper also proposes the Visual Needle-in-a-Haystack with Distractors benchmark to evaluate the retrieval capability. This paper follows the commonly used metrics like average accuracy to compare again baselines.

**Other Comments Or Suggestions:**

The authors are encouraged to address the concerns in the weakness section.

**Other Strengths And Weaknesses:**

Strengths:

-The proposed VideoRoPE is novel to the best my my knowledge.

-The improvement on many video benchmarks is significant.

-The paper is well-structured and easy to follow. The problem is clearly defined, the proposed method is clearly explained, and the experimental results are clearly presented.

Weaknesses:

-Only one LLM i.e., QWen2-VL, is evaluated. It would be more convincing to show improvement over more base LLMs.

-More recent (SOTA) baselines should be compared. For example, LLaVA-Video, LLaVA-OneVision with different sizes.

-The performance degradation at 64k context length (Tab. 2) is not well explored in the main paper.

**Questions For Authors:**

See the weakness.

**Relation To Broader Scientific Literature:**

This paper proposes VideoRoPE, an new video positional embedding that is aware of spatiotemporal information. Compared to existing RoPE variants (Vanilla RoPE, TAD-RoPE, and M-RoPE), it has nice properties such as 2D/3D structure awareness, frequency allocation, spatial symmetry, and temporal index scaling. Overall, I believe this paper provides a valuable contribution to the field of video understanding with LLMs.

**Theoretical Claims:**

There is no formal proof and no theoretical claim in the paper.

---

> ### Author Rebuttal · Authors · 2025-04-01
>
> Dear Reviewer t9dP,
>
> Thanks for your valuable feedback. We sincerely thank all reviewers for acknowledging that this paper is novel (t9dP, juMQ),  well-motivated (t9dP, 26sL, 3qrY), significant improvements (t9dP, 26sL, 3qrY), well-written (t9dP, juMQ, 26sL, 3qrY), and easy to follow (t9dP, 3qrY). Below we address your questions:
>
> > Q1: Only one LLM i.e., QWen2-VL, is evaluated. It would be more convincing to show improvement over more base LLMs.
>
> Thanks for your suggestion. We have updated the table below with results on QWen2.5-VL. Performance is reported across three categories: long video understanding (LongVideoBench, MLVU, VideoMME) with a 64k context length, short video hallucination at an 8k context length, and average retrieval accuracy. Across all six benchmarks, VideoRoPE's performance remains consistently superior to other RoPE variants, including Vanilla RoPE, TAD-RoPE, and M-RoPE. We will add these results to our revised version.
>
> |Qwen2.5VL-7B|LongVideoBench|MLVU|VideoMME|VideoHallucer|V-NIAH|V-NIAH-D|
> |-|-|-|-|-|-|-|
> |Vanilla RoPE|53.37|63.13|60.0|45.1|31.77|29.99|
> |TAD-RoPE|54.62|66.78|59.0|45.1|28.44|24.88|
> |M-RoPE|58.71|68.09|60.6|45.3|77.33|75.11|
> |VideoRoPE|**59.20**|**68.84**|**61.6**|**45.8**|**86.44**|**86.22**|
>
> > Q2: More recent (SOTA) baselines should be compared. For example, LLaVA-Video, LLaVA-OneVision with different sizes.
>
> We appreciate your feedback. We have added the experimental results of Video-RoPE with the SOTA models including LLaVA OneVision, LongVU, Apollo and LLaVA-Video with different sizes (3B and 7B).
>
> Results with 3B size models:
> |Model|LongVideoBench|MLVU|VideoMME|
> |-|-|-|-|
> |VILA1.5-3B|42.9|44.4|42.2|
> |Phi-3.5-Vision-4.2B|-|-|50.8|
> |LongVU-3.2B|-|55.9|51.5|
> |Qwen2.5-VL-3B-VideoRoPE|**54.7**|**62.6**|**58.3**|
>
> Results with 7B size models:
> |Model|LongVideoBench|MLVU|VideoMME|
> |-|-|-|-|
> |LLaVA OneVision-7B|56.3|64.7|58.2|
> |LongVU-7B|-|65.4|60.6|
> |Apollo-7B|58.5|**70.9**|61.3|
> |LLaVA-Video-7B|58.2|70.8|63.3|
> |Qwen2.5-VL-7B-VideoRoPE|**62.0**|70.7|**64.4**|
>
> Notably, our method achieves better results with significantly less fine-tuning data (0.33 million) than prior SOTA approaches (8.8 million for LLaVA OneVision, 3.75 million for LongVU, 3.2 million for Apollo, and 2.7 million for LLaVA-Video). Our results highlight the benefits of using our RoPE design in Video Large Language Models.
>
> > Q3: The performance degradation at 64k context length (Tab. 2) is not well explored in the main paper.
>
> In the Tab. 2 of main paper, the performance of 64k (65.56 for MLVU, 61.33 for VideoMME) is slightly worse that 32k (66.02 for MLVU, 61.67 for Video-MME) on some benchmarks. The reason is that simply increasing the context window doesn't guarantee improved performance. It is challenging for the model's ability to effectively process and utilize extremely long contexts. For example, in Figure 5 of the technical report of QWen2-VL based on M-RoPE, the performance of 64k (70.4) is also slightly worse than 48k (71.3). We further tested the performance on our VideoRoPE on 48k and 80k context length. The results at 48k and 80k showed performance fluctuations, which are consistent with Figure 5 of the QWen2-VL technical report.
>
> |Method (VideoRoPE)|8k|16k|32k|48k|64k|80k|
> |-|-|-|-|-|-|-|
> |LongVideoBench|54.46|55.29|57.15|56.22|57.26|56.53|
> |MLVU|65.19|66.29|66.02|66.43|65.56|64.88|

---

> > ### Comment · Reviewer_t9dP · 2025-04-09
> >
> > Thanks for providing the rebuttal. I would keep my original rating. I do not improve my rating because the new results are not completely convincing.
> >
> > - Regarding the experiments with QWen2.5-VL, there is only marginal improvement over M-RoPE (58.71 vs 59.2 on LongVideoBench, 68.09 vs 68.84 on MLVU, 45.3 vs 45.8 on VideoHallucer). It seems that the improvement of VideoRoPE saturates with a stronger base LLM.
> > - Why don't the numbers of Qwen2.5-VL-7B-VideoRoPE in the third table match the numbers of the first table? The first table also shows the results with Qwen2.5-VL-7B, right?

---

> > > ### Author Response · Authors · 2025-04-09
> > >
> > > Dear Reviewer t9dP,
> > >
> > > We understand the high workload of reviewers, and your comments come when there is only one minute left (April 8th, AoE) for the author's rebuttal. Therefore, to ensure you still have time to respond, we managed to submit the rebuttal above quickly. Please kindly take a look at our rebuttal and don't hesitate to respond with more thoughts.
> > >
> > > > Q1: Regarding the experiments with QWen2.5-VL, there is only marginal improvement over M-RoPE (58.71 vs 59.2 on LongVideoBench, 68.09 vs 68.84 on MLVU, 45.3 vs 45.8 on VideoHallucer). It seems that the improvement of VideoRoPE saturates with a stronger base LLM.
> > >
> > > We appreciate the reviewer’s observation that the performance gains of VideoRoPE over M-RoPE are relatively smaller on some benchmarks when applied to Qwen2.5-VL. Nonetheless, VideoRoPE still consistently brings improvements across **all six benchmarks**, expecially on retrieval benchmarks (77.33 -> 86.44 on V-NIAH, 75.11 -> 86.22 on V-NIAH-D), which we believe demonstrates its robustness and general applicability, even on recent SOTA vision-language models. Furthermore, our goal is to provide a general method that enhances long-context video understanding, and consistent improvements, albeit marginal in some cases, support the effectiveness of our approach.
> > >
> > > > Q2: Why don't the numbers of Qwen2.5-VL-7B-VideoRoPE in the third table match the numbers of the first table? The first table also shows the results with Qwen2.5-VL-7B, right?
> > >
> > > That's because **we report the results under different settings**. The caption of the first table already indicates that the results are based on a 64k context, which is intended to demonstrate the model’s long-context reasoning ability. As for the third table, since it is a comparison with SOTA models, we report the best results across different context lengths for each model.
> > >
> > > We have already addressed all the concerns and with detailed response. We are actively looking forward to your response and further feedback.

---

### Decision · Program_Chairs · 2025-05-01

**Decision:**

Accept (oral)

**Comment:**

This paper received three accept recommendations and one strong accept, reflecting a positive consensus among reviewers. The work is well-written, with a clearly motivated problem and a methodologically sound approach to video position embedding. Empirical results show consistent improvements across multiple tasks, supporting the effectiveness of the proposed method. While the paper is generally clear and well-organized, reviewers noted minor issues related to typos, awkward phrasing, and a few claims lacking sufficient supporting evidence. The authors are advised to address these presentation and clarity concerns in the final version to strengthen the overall impact of the paper.